# Causal Effect Estimation with Mixed Latent Confounders and Post-treatment Variables

## Abstract

Causal inference from observational data has attracted considerable attention among researchers. One main obstacle is the handling of confounders. As direct measurement of confounders may not be feasible, recent methods seek to address the confounding bias via proxy variables, i.e., covariates postulated to be conducive to the inference of latent confounders. However, the selected proxies may scramble both confounders and post-treatment variables in practice, which risks biasing the estimation by controlling for variables affected by the treatment. In this paper, we systematically investigate the bias due to latent post-treatment variables, i.e., ***latent post-treatment bias***, in causal effect estimation. Specifically, we first derive the bias when selected proxies scramble both confounders and post-treatment variables, which we demonstrate can be arbitrarily bad. We then propose a novel Confounder-identifiable VAE (CiVAE) to address the bias. Based on a mild assumption that the prior of latent variables that generate the proxy belongs to a general exponential family with at least one invertible sufficient statistic in the factorized part, CiVAE *individually* identifies latent confounders and latent post-treatment variables up to bijective transformations. We then prove that with individual identification, the intractable disentanglement problem of latent confounders and post-treatment variables can be transformed into a tractable independence test problem. Finally, we prove that the true causal effects can be unbiasedly estimated with transformed confounders inferred by CiVAE. Experiments on both simulated and real-world datasets demonstrate significantly improved robustness of CiVAE.

## 1 Introduction

Causal inference, which aims to infer cause-and-effect relations from data, has gained increasing prominence in various fields, such as social science, economics, and public health [10, 17, 34]. Traditional methods rely on the golden standard of randomized control trials (RCT) to draw valid causal conclusions via experimentation [6]. Recently, more attention has been dedicated to causal inference from observational data, where treatments, outcomes, and unit features are passively observed, and researchers have no control over the treatment assignment mechanism [36, 37, 40].

One main obstacle to inferring valid causal relations from observational data is the confounding bias, which occurs when we fail to account for the systematic difference between the treatment and non-treatment group due to variables that causally influence the past treatments and the outcome, i.e., unobserved confounders [16]. If the confounders can be measured, a simple strategy to address the bias is to control them via covariate adjustment [33] or propensity score re-weighting [24]. However, confounders are not always measurable [23]. Therefore, recent methods seek to adjust for the influence of unobserved confounders based on their proxies, which are easily acquirable covariates postulated to be causally related with the unobserved confounders [29, 42, 28]. One exemplar work

Submitted to 38th Conference on Neural Information Processing Systems (NeurIPS 2024). Do not distribute.

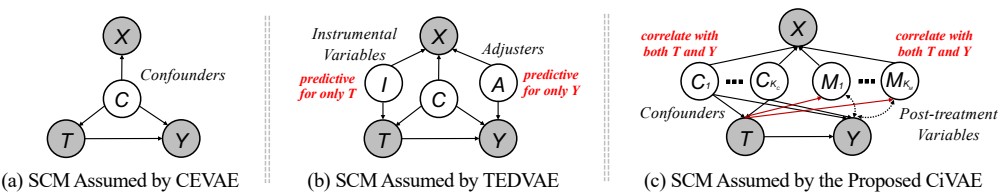

Figure 1: Comparison between the causal models assumed by CEVAE, TEDVAE, and CiVAE.

is the causal effect variational auto-encoder (CEVAE) [25], which has demonstrated that confounding bias can be mitigated by controlling latent variables inferred from the proxies of confounders.

Although proxy-based methods have achieved substantial progress in recent years, they may risk controlling latent post-treatment variables scrambled in the proxies, where **latent post-treatment bias** can be introduced. Here, we note that the negative effects of controlling *observed* post-treatment variables have been investigated in prior research [1, 9, 21]. For example, Montgomery et al. [30] found that more than 50% of the papers published in top journals of politics *inadvertently control post-treatment variables* in the experimental setting, even though researchers have complete control over which covariates to control for. On this basis, we postulate that the post-treatment bias could be even worse for proxy-based methods in the setting of observational study where variables are passively recorded. In addition, the post-treatment variables can be **latent** and scrambled into the observed covariates together with the latent confounders, which makes them difficult to disentangle.

Consider a real-world example from the Company[1]. We found that *changing* a job from onsite to online mode causes applicants to make different decisions, and we want to estimate the causal effects of *switching a job from onsite to online mode* to *the decisions of the applicants* (reflected by statistics of applicants that apply for the job). In this case, the Company collected two groups of online (treated) and onsite (control) jobs, where the statistics of the applicants (e.g., the average age) are calculated as the surrogate outcome. Clearly, job seniority is a confounder, since less senior jobs are more likely to permit online work, and applicants for these jobs tend to be younger. However, the seniority level of a job can be difficult to measure. Therefore, the required skills of the job can be used as the proxy of the confounder "seniority", as senior jobs tend to require more advanced skills. However, **a caveat** is that switching to an online work mode may also alter the required skills of a job, thereby affecting the qualification and, therefore, the decision of the applicants. Consequently, directly using the skills as the proxy of the confounder "seniority" for adjustment could unintentionally control latent mediators (changed skills), which introduces latent post-treatment bias in the causal effect estimation.

Addressing the **latent post-treatment bias** faces multi-faceted challenges. First, there lacks a theoretical formulation of the bias when selected proxies scramble latent post-treatment variables for existing proxy-based methods. In addition, it is difficult to distinguish confounders and post-treatment variables in the latent space due to their similar observed behaviors. Existing covariate disentanglement-based methods, e.g., TEDVAE [44], focus on an easier task of disentangling latent confounders with latent adjusters and instrumental variables, which can be achieved by leveraging their different predictive abilities w.r.t. the treatment and outcome. However, since both latent confounders and post-treatment variables correlate with the treatment and the outcome, they cannot be disentangled by these methods. Finally, even if latent confounders can be distinguished from post-treatment variables, since most existing latent variable models have no identifiability guarantee [19], it is unclear whether controlling the inferred latent variables, which may be arbitrary transformations of the true confounders, can provide unbiased estimations of true causal effects.

To address the aforementioned challenges, we first analyze existing proxy-based methods when selected proxies scramble both latent confounders and post-treatment variables and show the estimation can be arbitrarily biased. We then propose a novel Confounder-identifiable VAE (CiVAE) to address the latent post-treatment bias. Specifically, we prove that based on a mild assumption that the prior of latent variables that generate the observed proxy (i.e., the latent confounders and post-treatment variables) belong to a general exponential family with at least one invertible sufficient statistic in the factorized part, latent confounders and latent post-treatment variables can be *individually* identified up to *simple bijective transformations*. With such identifiability guarantee, based on the causal relations among confounders, mediators, and treatment, we further demonstrate that the inferred confounders

---

[1]Anonymized due to double-blind review policy.

(which are actually transformed proxies of the true confounders) could be properly distinguished from the latent post-treatment variables with pair-wise conditional independence tests. Finally, we prove that the true causal effects can be unbiasedly estimated based on transformed confounders inferred by CiVAE. Experiments on both simulated and real-world datasets demonstrate that CiVAE shows more robustness to latent post-treatment bias than existing methods.

## 2 Problem Formulation

In this paper, we assume the causal model in Fig. 1-(c). We use a binary random variable $T$ to denote the treatment, a random vector $\boldsymbol{X} \in \mathbb{R}^{K_X}$ to denote the observed covariates (i.e., the proxy), and a random scalar $Y \in \mathbb{R}$ to denote the outcome. Furthermore, the observed covariates $X$ are assumed to be generated from $K_C$ independent latent confounders $\boldsymbol{C} \triangleq [C_1, C_2..., C_{K_C}]$ causally influencing both $T$ and $Y$, and $K_M$ latent post-treatment variables $\boldsymbol{M} \triangleq [M_1, M_2..., M_{K_M}]$ under the causal influence of the treatment (where the relation between $\boldsymbol{M}$ and $Y$ can be arbitrary). We use the random vector $\boldsymbol{Z} \triangleq [\boldsymbol{C}||\boldsymbol{M}] \in \mathbb{R}^{K_Z=K_C+K_M}$ to denote all latent factors. **Our aim** is to estimate the average causal effects of treatment $T$ on outcome $Y$ with auxiliary confounder information in $\boldsymbol{X}$, where the estimation should be devoid of both confounding bias and post-treatment bias.

## 3 Theoretical Analysis of Latent Post-Treatment Bias

### 3.1 Preliminaries and Assumptions

To achieve such a purpose, we first define the (conditional) average treatment effects (C/ATE) when covariates $\boldsymbol{X}$ scramble both latent confounders $\boldsymbol{C}$ and post-treatment variables $\boldsymbol{M}$. We then define the post-treatment bias when covariates $\boldsymbol{X}$ are directly used as the proxy of confounders. To facilitate the analysis, we make the following assumption regarding the causal generative process.

**Assumption 1.** *(**Noisy-Injectivity**). We assume $\boldsymbol{X} = f(\boldsymbol{C}, \boldsymbol{M}) + \epsilon$, where $f$ is a deterministic function that combines latent confounders $\boldsymbol{C}$ and latent post-treatment variables $\boldsymbol{M}$ into observations $\boldsymbol{X}$, and $\epsilon$ is random noise. In addition, we assume that the function $f$ is **injective**; beyond injectivity, $f$ can be arbitrarily nonlinear. We use $f^\dagger : \boldsymbol{X} \to [\boldsymbol{C}||\boldsymbol{M}]$ to denote its left inverse. We use $f_C^\dagger : \boldsymbol{X} \to \boldsymbol{C}$ and $f_M^\dagger : \boldsymbol{X} \to \boldsymbol{M}$ to denote the mapping from $\boldsymbol{X}$ to $\boldsymbol{C}$, $\boldsymbol{M}$, respectively.*

*Noisy-Injectivity* is a common assumption made either explicitly or implicitly in most existing proxy-of-confounder-based causal inference algorithms. For example, if both $\boldsymbol{X}$ and $\boldsymbol{C}$ are categorical, [31] assumes that $\boldsymbol{X}$ has at least the same number of categories as $\boldsymbol{C}$, whereas the effect restoration algorithm [35] assumes that the matrix of $p(\boldsymbol{C}, \boldsymbol{X})$ to be full-rank. Although CEVAE [25] makes no explicit injectivity assumption between $\boldsymbol{C}$ and $\boldsymbol{X}$, it requires that the joint distribution $p(\boldsymbol{C}, \boldsymbol{X}, T, Y)$ can be fully recovered from the observations $(\boldsymbol{X}, T, Y)$. [2] show that some of the possible identification criteria for the recovery include **1)** having multiple independent views of $\boldsymbol{C}$ in $\boldsymbol{X}$ [8], and **2)** $\boldsymbol{C}$ is categorical and $\boldsymbol{X}$ is a mixture of Gaussian components determined by $\boldsymbol{C}$ (that is, $\boldsymbol{X}$ is generated by bijective mapping of $\boldsymbol{C}$ to the mean of the corresponding component with added Gaussian noise).

In the following part of this section, we omit the noise $\epsilon$ to gain better intuition of latent post-treatment bias (but all the exact conclusions will still hold in the posterior sense [19]). In Section 4, we assume noise exists and demonstrate that our method can still properly identify the latent confounders.

### 3.2 Causal Estimand and the True ATE

Based on Assumption 1, we are ready to define the estimand of average treatment effect (ATE) through controlling the covariates $\boldsymbol{X}'$, as well the as the true (conditional) average treatment effects.

**Definition 1.** *(**DCEV & DEV**). We define the Difference in Conditional Expected Values (DCEV) as:*

$$DCEV(\boldsymbol{x}') = \mathbb{E}[Y|T=1, \boldsymbol{X}'=\boldsymbol{x}'] - \mathbb{E}[Y|T=0, \boldsymbol{X}'=\boldsymbol{x}'], \tag{1}$$

*which is the difference of the expected value of $Y$ for units with variable $\boldsymbol{X}' = \boldsymbol{x}'$ in the treatment group and the non-treatment group. Based on $DCEV(\boldsymbol{x}')$, we define the Difference in Expected Value (DEV) as $DEV(\boldsymbol{X}') = \mathbb{E}_{p(\boldsymbol{X}')}[DCEV(\boldsymbol{X}')]$ as the expectation of $DCEV$ w.r.t. $p(\boldsymbol{X}')$.*

$DEV(\boldsymbol{X}')$ denotes the estimand of ATE when $\boldsymbol{X}'$ is the covariates that we choose to control (i.e., calculate the expected difference in each stratum of $\boldsymbol{X}' = \boldsymbol{x}'$). If $\boldsymbol{X}' = \emptyset$, $DEV(\emptyset)$ represents the *naive estimator* that directly calculates the expected difference of the outcome $Y$ between the treatment group and the non-treatment group. With the causal estimand $DEV(\boldsymbol{X}')$ defined, we then derive the true causal effects with the covariates $\boldsymbol{X}'$ when it scrambles both latent confounders and post-treatment variables according to the generative process described in Assumption 1:

**Definition 2.** *Under Assumption 1, we define the Conditional Average Treatment Effect (CATE) for individuals with observed covariates $\boldsymbol{X} = \boldsymbol{x}$ by controlling only the confounder part in $\boldsymbol{X}$ as:*

$$CATE(\boldsymbol{x}) = \mathbb{E}[Y|T=1, \boldsymbol{C} = f_C^\dagger(\boldsymbol{x})] - \mathbb{E}[Y|T=0, \boldsymbol{C} = f_C^\dagger(\boldsymbol{x})], \tag{2}$$

*with the Average Treatment Effect (ATE) of treatment $T$ defined as:*

$$ATE = \mathbb{E}[Y|do(T=1)] - \mathbb{E}[Y|do(T=0)] = \mathbb{E}_{p(\boldsymbol{C})}[\mathbb{E}[Y|T=1, \boldsymbol{C}] - \mathbb{E}[Y|T=0, \boldsymbol{C}]]. \tag{3}$$

Please note that we only consider the latent confounder component of the observed features $\boldsymbol{X}$ in the definition of CATE in Eq. (2). This is because the causal relationship between the post-treatment variables $\boldsymbol{M}$ and the outcome $Y$ is indeterminate. However, if the specific relationship between $\boldsymbol{M}$ and $Y$ can be further established by the researcher (e.g., all elements of $\boldsymbol{M}$ are latent mediators), more precise forms of CATE can be derived with path-specific counterfactual analysis [5, 14].

### 3.3 Latent Post-Treatment Bias

With $DEV(\boldsymbol{X}')$ (the ATE estimator that control for the covariates $\boldsymbol{X}'$), CATE, and ATE defined in Section 3.2, in this section, we analyze the *latent post-treatment bias* of existing proxy-of-confounder-based causal inference methods, such as CEVAE, that control for latent variables inferred from the covariates $\boldsymbol{X}$ to estimate the ATE of $T$ on $Y$, when $\boldsymbol{X}$ scrambles both latent confounders and post-treatment variables as Assumption 1. In our analysis, Lemma 3.1 will be frequently used.

**Lemma 3.1.** *For an injective function $g$, $\mathbb{E}[Y|\boldsymbol{X}' = \boldsymbol{x}'] = \mathbb{E}[Y|g(\boldsymbol{X}') = g(\boldsymbol{x}')]$ holds.*

The proof when $g$ is differentiable *a.e.* can be referred to in Appendix C.1. Since the latent variable models used in existing methods (such as VAE with factorized Gaussian prior in CEVAE) lack identifiability guarantee (i.e., the recovery of the exact latent variables), we assume that these models can recover the true latent space $\boldsymbol{Z} = [\boldsymbol{C}, \boldsymbol{M}]$ up to invertible transformations $\bar{f}$, where the inference process can be represented as $\hat{\boldsymbol{Z}} = \tilde{f}(\boldsymbol{X}) = \bar{f} \circ f^\dagger(\boldsymbol{X})$. With such an assumption, we have the following theorem regarding the latent post-treatment bias when $\boldsymbol{X}$ mixes post-treatment variables.

**Theorem 3.2.** *If the observed covariates $\boldsymbol{X}$ are generated from latent confounders $\boldsymbol{C}$ and latent post-treatment variables $\boldsymbol{M}$ according to Assumption 1, the latent post-treatment bias of a proxy-based causal inference algorithm that controls latent variables $\hat{\boldsymbol{Z}}$ inferred from $\boldsymbol{X}$ via $\tilde{f} = \bar{f} \circ f^\dagger$ : $\mathbb{R}^{K_X} \to \mathbb{R}^{K_C + K_M}$ to estimate the ATE can be formulated as follows:*

$$\begin{aligned} Bias(\boldsymbol{X}) &= ATE - DEV(\tilde{f}(\boldsymbol{X})) = ATE - \mathbb{E}[\mathbb{E}[Y|T=1, \tilde{f}(\boldsymbol{X})] - \mathbb{E}[Y|T=0, \tilde{f}(\boldsymbol{X})]] \\ &= ATE - \mathbb{E}[\mathbb{E}[Y|1, \bar{f} \circ f^\dagger(f(\boldsymbol{C}, \boldsymbol{M}))] - \mathbb{E}[Y|0, \bar{f} \circ f^\dagger(f(\boldsymbol{C}, \boldsymbol{M}))]] \\ &= \mathbb{E}[\mathbb{E}[Y|1, \boldsymbol{C}] - \mathbb{E}[Y|0, \boldsymbol{C}]] - \mathbb{E}[\mathbb{E}[Y|1, \boldsymbol{C}, \boldsymbol{M}] - \mathbb{E}[Y|0, \boldsymbol{C}, \boldsymbol{M}]], \end{aligned} \tag{4}$$

*which can be arbitrarily bad. Therefore, the estimator of existing proxy-of-confounder-based methods, i.e., $DEV(\tilde{f}(\boldsymbol{X}))$, is an arbitrarily biased estimator of the ATE, when the selected proxy of confounders $\boldsymbol{X}$ accidentally mixes in latent post-treatment variables $\boldsymbol{M}$.*

The final step of Eq. (4) can be proved since $f$ is injective and $\bar{f}$ bijective, the composite $\bar{f} \circ f^\dagger \circ f$ : $[\boldsymbol{C}, \boldsymbol{M}] \to \hat{\boldsymbol{Z}}$ is bijective, so we can use Lemma 3.1 to remove $\bar{f} \circ f^\dagger \circ f$ in the condition.

### 3.4 Examples in the Linear Case

Generally, the latent post-treatment bias defined in Eq. (4) cannot be simplified, because *(i)* the causal relationship between $\boldsymbol{M}$ and $Y$ are indeterminate, and *(ii)* the causal influence of $\boldsymbol{C}$, $\boldsymbol{M}$, and $T$ on $Y$ can be arbitrary. However, for linear structural causal models with determined causal relationships between $\boldsymbol{M}$ and $Y$ (e.g., $\boldsymbol{M}$ are mediators, which are post-treatment variables that have causal influences on the outcomes), stronger conclusions can be drawn as follows:

**Corollary 3.3.** *(Mixed Latent Mediator). For the linear Structural Causal Model (SCM) defined as:*

$$(i) \ T \leftarrow \mathbb{1}(\alpha_T + \sum \beta_i \cdot C_i > a), \ (ii) \ M_j \leftarrow \alpha_M + \gamma_j \cdot T$$
$$(iii) \ \boldsymbol{X} \leftarrow \boldsymbol{\alpha}_X + \mathbf{A}[\boldsymbol{M}||\boldsymbol{C}], \ (iv) \ Y \leftarrow \alpha_Y + \tau \cdot T + \sum \theta_j \cdot M_j + \sum \kappa_i \cdot C_i, \quad (5)$$

*where the mixture function $f = \mathbf{A} \in \mathbb{R}^{K_X \times (K_C + K_M)}$ is a full column-rank matrix, the CATE, ATE, and the bias of proxy-of-confounder-based causal inference model that controls the latent variables $\hat{\boldsymbol{Z}}$ inferred via $\hat{\boldsymbol{Z}} = \tilde{f}(\boldsymbol{X}) = \mathbf{B}^T \boldsymbol{X}$ can be formulated as follows:*

$$ATE = CATE = \tau + \sum \gamma_j \cdot \theta_j, \ \text{and } DEV(\hat{\boldsymbol{Z}}) = \mathbb{E}[DCEV(\hat{\boldsymbol{Z}})] = DCEV(\hat{\boldsymbol{Z}}) = \tau$$
$$Bias(\hat{\boldsymbol{Z}}) = ATE - DEV(\hat{\boldsymbol{Z}}) = \sum \gamma_j \cdot \theta_j, \quad (6)$$

*where $\mathbf{B} \in \mathbb{R}^{K_X \times (K_C + K_M)}$ is another full column-rank matrix. Since $\sum \gamma_j \cdot \theta_j$ is arbitrary, the estimator $DEV(\hat{\boldsymbol{Z}}) = \mathbb{E}[DCEV(\mathbf{B}^T \boldsymbol{X})]$ is arbitrarily biased for ATE estimation.*

The proof of Eq. (6) is provided in Appendix C.2. In addition, we show that post-treatment variables $\boldsymbol{M}$ DO NOT necessarily need to have direct causal effects on the outcome $Y$ to incur arbitrary bias in ATE estimation. In Appendix C.3, we provide another example (i.e., Mixed Latent Correlator) in the linear case where $\boldsymbol{M}$ is correlated with $Y$ through unobserved confounders $\boldsymbol{U}$ in Corollary C.1.

## 4 Methodology

In this section, we introduce the proposed Confounder-identifiable Variational Auto-Encoder (**CiVAE**) in detail. Specifically, we first prove that if the prior distribution of the true latent variables $\boldsymbol{Z} = [\boldsymbol{C}, \boldsymbol{M}]$ satisfies certain weak assumptions, CiVAE *individually* identify $[\boldsymbol{C}, \boldsymbol{M}]$ up to bijective transformations. Then, utilizing the causal relations between $\boldsymbol{C}$, $\boldsymbol{M}$, and $T$, we novelly transform the challenging confounder-identifiability problem into a tractable pair-wise conditional independence test problem, which can be effectively solved with kernel-based methods. The generalization of CiVAE to address the interactions among $[\boldsymbol{C}, \boldsymbol{M}]$ are discussed in Section D of the Appendix.

### 4.1 Generative Process

The fundamental work on the identifiability of deep variational inference, i.e., the identifiable VAE (iVAE) [19], makes a strict assumption that the prior of true latent variables $\mathbf{Z}$ (i.e., $[\boldsymbol{C}, \boldsymbol{M}]$ in our case) is conditionally factorized given the available covariates. However, since both $\boldsymbol{C}$ and $\boldsymbol{M}$ form fork structures with the outcome $Y$ (see Fig. 1-(c)) [22], $C_i, C_j, M_i$, and $M_j$ are not independent given $Y$. Recently, Non-Factorized iVAE (NF-iVAE) [26] was proposed that allows arbitrary dependence among the true latent variables $\boldsymbol{Z}$ in the conditional priors, where $\boldsymbol{Z}$ can be identified up to arbitrary non-linear transformations. However, the transformation is not necessarily invertible, which is risky as multiple values of the confounders may collapse, leading to bias when estimating the ATE by averaging the $DCEV$ calculated in each stratum of the inferred confounders.

In contrast to NF-iVAE, CiVAE guarantees the individual and bijective identifiability of $\boldsymbol{Z}$ by putting a general exponential family *with at least one invertible sufficient statistic in the factorized part* as its prior when conditioning on treatment $T$ and outcome $Y$, which can be formulated as follows.

**Assumption 2.** *Let $\boldsymbol{Z} = [\boldsymbol{C}||\boldsymbol{M}]$ be the random vector for latent variables that causally generate the observed covariates $\boldsymbol{X}$ according to Assumption 1. We assume that the conditional prior of $\boldsymbol{Z}$ given the outcome $Y$ and the treatment $T$ belongs to a general exponential family with parameter vector $\boldsymbol{\lambda}(Y, T)$ and sufficient statistics $\boldsymbol{S}(\boldsymbol{Z}) = [\boldsymbol{S}_f(\boldsymbol{Z})^T, \boldsymbol{S}_{nf}(\boldsymbol{Z})^T]^T$. Specifically, $\boldsymbol{S}(\boldsymbol{Z})$ is composed of (i) the sufficient statistics of a factorized exponential family, i.e., $\boldsymbol{S}_f(\boldsymbol{Z}) = [\boldsymbol{S}_1(Z_1)^T, \cdots, \boldsymbol{S}_{K_Z}(Z_{K_Z})^T]^T$, where all components $\boldsymbol{S}_i(Z_i)$ have dimension larger than or equal to 2 and **each $\boldsymbol{S}_i$ has at least one invertible dimension**, and (ii) $\boldsymbol{S}_{nf}(\boldsymbol{Z})$, where $\boldsymbol{S}_{nf}$ is a neural network with ReLU activation. The density of the conditional prior can be formulated as:*

$$p_{\boldsymbol{S}, \boldsymbol{\lambda}}(\boldsymbol{Z}|Y, T) = \mathcal{Q}(\boldsymbol{Z})/\mathcal{C}(Y, T) \exp[\boldsymbol{S}(\boldsymbol{Z})^T \boldsymbol{\lambda}(Y, T)], \quad (7)$$

*where $\mathcal{Q}(\boldsymbol{Z})$ is the base measure, and $\mathcal{C}(Y, T)$ is the normalizing constant independent of $\boldsymbol{Z}$.*

We justify that assumption 2 is weak and practical as follows. *(i)* Neural networks with ReLU activation have **universal approximation ability** of distributions [27]. Therefore, Eq. (7) can model arbitrary dependence between true latent confounders $C$ and post-treatment variables $M$ conditional on $T$ and $Y$. *(ii)* Although CiVAE makes an extra assumption that $\forall i$, at least one dimension of $S_i$ is invertible, this can be easily satisfied as most commonly used exponential family distributions, such as Gaussian, Bernoulli, etc., has at least one invertible sufficient statistics[2].

The reason why we use ReLU as the activation is that, the identifiability of iVAE relies on the condition that the sufficient statistics $S$ have zero second-order cross-derivative. The factorized part, i.e., $S_f$, satisfies it trivially as all cross-derivatives of $S_f$ are zero. In addition, since the ReLU neural networks are linear *a.e.*, all second-order derivatives of $S_{nf}$ are zero. Therefore, identifiability holds after adding $S_{nf}$ in the prior that allows the capturing of arbitrary dependence among $Z$.

## 4.2 Optimization Objective

Combining Assumptions 1 and 2, the generative process assumed by CiVAE can be formulated as:

$$(i)\, p_{\boldsymbol{\theta}}(\boldsymbol{X}, \boldsymbol{Z} \mid Y, T) = p_f(\boldsymbol{X} \mid \boldsymbol{Z}), (ii)\ p_{\boldsymbol{S}, \boldsymbol{\lambda}}(\boldsymbol{Z} \mid Y, T),\ (iii)\, p_f(\boldsymbol{X} \mid \boldsymbol{Z}) = p_{\boldsymbol{\epsilon}}(\boldsymbol{X} - f(\boldsymbol{Z})). \quad (8)$$

where $\boldsymbol{\theta} = (f, \boldsymbol{\lambda}, \boldsymbol{S}) \in \Theta$ are the parameters of the generative distribution. Since the generative process of CiVAE is parameterized by deep neural networks, the posterior distribution of $\boldsymbol{Z}$, i.e., $p_{\boldsymbol{\theta}}(\boldsymbol{Z} \mid \boldsymbol{X}, Y, T)$, is intractable. Therefore, we resort to variational inference [4], where we introduce an approximate posterior $q_{\boldsymbol{\phi}}(\boldsymbol{Z} \mid \boldsymbol{X}, Y, T)$ parameterized by a deep neural network with a trainable parameter $\boldsymbol{\phi}$, and in $q_{\boldsymbol{\phi}}(\boldsymbol{Z}|\cdot)$ finds the one closest to $p_{\boldsymbol{\theta}}(\boldsymbol{Z}|\cdot)$ measured by KL divergence. The minimization of KL is equivalent to maximization of the evidence lower bound (ELBO):

$$\mathcal{L}(\boldsymbol{\theta}, \boldsymbol{\phi}) := \mathbb{E}_{q_{\boldsymbol{\phi}}} \big[ \log p_f(\boldsymbol{X} \mid \boldsymbol{Z}) + \underbrace{\log p_{\boldsymbol{S}, \boldsymbol{\lambda}}(\boldsymbol{Z} \mid Y, T) - \log q_{\boldsymbol{\phi}}(\boldsymbol{Z} \mid \cdot)}_{\text{KL of posterior with prior}} \big]. \quad (9)$$

Since the normalization constant $\mathcal{C}$ in Eq. (7) is generally intractable, it is infeasible to directly learn $\boldsymbol{S}, \boldsymbol{\lambda}$ by optimizing Eq. (9). Therefore, we substitute the KL term in Eq. (9) with the widely-used score matching [13] to learn unnormalized densities instead as follows:

$$\mathcal{L}(\boldsymbol{S}, \boldsymbol{\lambda}, \boldsymbol{\phi}) := \mathbb{E}_{q_{\boldsymbol{\phi}}(\boldsymbol{Z}|\cdot)} \left[ \| \nabla_{\boldsymbol{Z}} \log q_{\boldsymbol{\phi}}(\boldsymbol{Z} \mid \cdot) - \nabla_{\boldsymbol{Z}} \log p_{\boldsymbol{S}, \boldsymbol{\lambda}}(\boldsymbol{Z} \mid Y, T) \|^2 \right]$$
$$= \mathbb{E}_{q_{\boldsymbol{\phi}}(\boldsymbol{Z}|\cdot)} \left[ \sum_{j=1}^{K_Z} \left[ \frac{\partial^2 p_{\boldsymbol{S}, \boldsymbol{\lambda}}(\boldsymbol{Z} \mid Y, T)}{\partial Z_j^2} + \frac{1}{2} \left( \frac{\partial p_{\boldsymbol{S}, \boldsymbol{\lambda}}(\boldsymbol{Z} \mid Y, T)}{\partial Z_j} \right)^2 \right] \right] + \text{ const}, \quad (10)$$

## 4.3 Identifiability of CiVAE

With the generative process and optimization objective of CiVAE discussed in previous sub-sections, we are ready to introduce the final assumption of CiVAE, which, combined with Assumptions 1 and 2, leads to the main Theorem of this paper, which states the identifiability of CiVAE.

**Assumption 3.** *Assume the following: (i) The set $\{\boldsymbol{X} \in \mathcal{X} | \phi(\boldsymbol{X}) = 0\}$ has measure zero, where $\phi$ is the characteristic function of the density $p_f$ in Eq. (8). (ii) The sufficient statistics, $S_i$ in $S_f$ are all twice differentiable. (iii) The mixture function $f$ in Eq. (8) has all second-order cross derivatives. (iv) There exist $k + 1$ distinct points $(Y, T)_0, \cdots, (Y, T)_k$ s.t. the matrix $\mathbf{L} = [\boldsymbol{\lambda}((Y, T)_1) - \boldsymbol{\lambda}((Y, T)_0), \cdots, \boldsymbol{\lambda}((Y, T)_k) - \boldsymbol{\lambda}((Y, T)_0)]$ of size $k \times k$ is invertible, where $k = Dim(\boldsymbol{S})$.*

Here, we note that Assumptions *(i) - (iii)* are trivial for differentiable neural networks. The Assumption *(iv)* can be intuitively understood as independent samples of $(Y, T)$ are required to identify $C$ and $M$. The identifiability theorem of CiVAE can be formulated as follows.

**Theorem 4.1.** *If Assumptions 1, 2, and 3 hold, and if $\boldsymbol{\theta}, \tilde{\boldsymbol{\theta}} \in \Theta \to p_{\boldsymbol{\theta}}(\boldsymbol{X}|Y, T) = p_{\tilde{\boldsymbol{\theta}}}(\boldsymbol{X}|Y, T)$, the true latent variables $\boldsymbol{Z}$ are identifiable up to **permutation** and **element-wise bijective transformation**. Furthermore, in the case of **variational inference**, if we denote the true parameter that generates the data as $\boldsymbol{\theta}^*$, if (i) the distribution family $q_{\boldsymbol{\phi}}(\boldsymbol{Z}|\boldsymbol{X}, Y, T)$ contains the posterior $p_{\boldsymbol{\theta}}(\boldsymbol{Z}|\boldsymbol{X}, Y, T)$, and $q_{\boldsymbol{\phi}}(\boldsymbol{Z}|\boldsymbol{X}, Y, T) > 0$, (ii) we optimize Eq. (4) w.r.t. both $\boldsymbol{\theta}, \boldsymbol{\phi}$, then in the limit of infinite data, true parameters $\boldsymbol{\theta}^*$ can be learned up to a permutation and bijective transformation of $\boldsymbol{Z}$.*

---

[2]There are a few exponential family dist. with no invertible sufficient statistics, e.g., Weibull with even shape parameter $k$. However, these distributions are not commonly used in statistics or machine learning.

The proof of Theorem 4.1 non trivially extends the NF-iVAE paper [26] by incorporating the new assumption introduced in CiVAE (i.e., each $S_i$ has at least one invertible dimension) to ensure that the transformation of each $Z_i$ is bijective. The detailed proof is provided in Appendix C.4 for reference.

### 4.4 Identification of Latent Confounders

Theorem 4.1 ensures that the latent variables $\hat{Z}$ inferred by CiVAE cannot *(i)* mix confounders and post-treatment variables in each dimension, or *(ii)* collapsing of different values of the latent confounders into the same value. To further determine the dimensions of confounder and post-treatment variable in $\hat{Z}$, we rely on the causal relations between latent variables $\hat{Z}$ and the treatment $T$ and the associated marginal/conditional dependence properties, which are discussed as follows.

- ***Case 1. Intra-Confounders.*** Latent confounders $C_i$, $C_j$ and the treatment $T$ form the *V structure* $C_i \to T \leftarrow C_j$. Therefore, $C_i$ and $C_j$ are marginally **independent**, whereas they become **dependent** when conditioning on the assigned treatment $T$.

- ***Case 2. Intra-Post Treatment Variables.*** Latent post-treatment variables $M_i$, $M_j$ and the treatment $T$ form a *Fork-structure* $M_i \leftarrow T \to M_j$, where $M_i$, $M_j$ are marginally **dependent**, but they become **independent** after conditioning on the assigned treatment $T$.

- ***Case 3. Cross-Confounder and Post-Treatment Variables.*** Latent confounder $C_i$, latent post-treatment variable $M_j$, and the treatment $T$ forms a Chain structure $C_i \to T \to M_j$, where $C_i$, $M_j$ are marginally dependent, and they become **independent** after conditioning on $T$.

From the above analysis we can find that, the dependence between two latent variables $\hat{Z}_i$ and $\hat{Z}_j$ **increases** after conditioning on the treatment $T$ ONLY in the case of *intra-confounders*. Therefore, if more than one latent confounder exists, which is highly probable when covariates $X$ are high-dimensional, we can conduct independence test $\texttt{Ind}(\hat{Z}_i, \hat{Z}_j)$ and $\texttt{CInd}(\hat{Z}_i, \hat{Z}_j | T)$ for all pairs of inferred latent variables, which can be implemented via kernel-based methods as [43], and select the pairs where the p-value of $\texttt{CInd}$ is larger than that of $\texttt{Ind}$ as latent confounders. Here, we note that the kernel-based (conditional) independence test incurs $N^2 \times K_Z^2$ complexity in the training phase. However, once the dimensions of the confounders in $\hat{Z}$ are determined, CiVAE **has the same complexity as CEVAE** for the estimation of CATE and ATE in the test phase.

### 4.5 ATE Estimator with Transformed Confounders

Finally, we demonstrate that controlling the transformed confounders $\hat{C}$ inferred by CiVAE provides an unbiased estimation of ATE. Specifically, we have the final Theorem show the unbiasedness.

**Theorem 4.2.** *Controlling bijective of confounders is equivalent to original confounders in ATE estimation, i.e., $DEV(\tilde{C}) = DEV(g(C)) = ATE$, if the transformation function $g$ is bijective.*

The proof of Theorem 4.2 for discrete $C$ is trivial (where $\hat{C} = g(C)$ represents a simple relabeling of the stratum that we calculate the $DCEV$ and take the expectation). The proof in the continuous case where $g$ is differentiable is provided in Appendix C.5. With Theorem 4.2, we can control the identified latent confounders as true confounders, providing an unbiased estimate of ATE.

## 5 Empirical Study

In this section, we provide and analyze the experiments we conduct on both simulated and real-world datasets, where a code demo written in PyTorch and Pyro is provided in this anonymous URL.

### 5.1 Datasets

**Simulated Datasets.** We first establish two simulated datasets, i.e., `LatentMediator` and `LatentCorrelator`, that consider two types of post-treatment variables, i.e., *(i)* mediators and *(ii)* correlators, i.e., variables that are correlated with the outcome $Y$ via latent confounders $U$, where the causal generative process is under the full control of the experimenter. The generative process of the two datasets can be referred to in Corollary 3.3 and Corollary C.1 in the Appendix, respectively. In our experiments, $C$ are generated from Gaussian distribution as $C \sim Gaussian(0, \mathbf{I}_{K_C})$. For

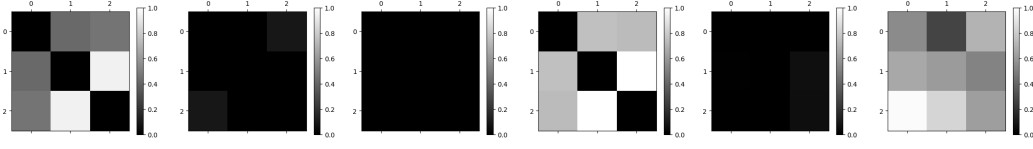

(a) **Case 1:** Intra-Confounder      (b) **Case 2:** Intra-Mediator      (c) **Case 3:** Confounder-Mediator

Figure 2: Visualization of $p$-value of independence test before and after conditioning on treatment $T$.

LatentMediator, $\boldsymbol{\gamma}$ is set as $[-1, -1, -1]$, $\boldsymbol{\theta}$ is set as $[1, 1, 1]$, and $\tau$ is set as 2, which results in $ATE = -1$. For the LatentCorrelator dataset, we set the same $\boldsymbol{\gamma}$ and $\boldsymbol{\theta}$ as the LatentMediator dataset, where parameters $\phi$ and $\tau$ are set to 1, which results in an overall $ATE$ of 1.

**Real-world Datasets.** In addition, we build real-world datasets from the Company to estimate the ATE of *switching a job from **onsite** to **online** work mode* to *the statistics of the applicants*. The average age and the variance of gender of the applicants are two outcomes of interest. Covariates $\boldsymbol{X} \in \{0, 1\}^{K_X}$ include the required skills of the job. Specifically, we establish a cohort of 3,228 jobs from the Bay Area in the US, where a preliminary study shows that $DEV(\emptyset) \approx 2$ years[3] (i.e., online job applicants are two years younger than onsite job applicants in the collected data), and $DEV(\emptyset) \approx -0.015$ (i.e., online jobs exhibit 0.015 more gender variance than onsite jobs in the collected data). To simulate $\boldsymbol{C}$ and $\boldsymbol{M}$, we first learn a generative model as follows:

$$\boldsymbol{Z} \sim Gaussian(\boldsymbol{0}, \mathbf{I}_{K_Z}), \boldsymbol{X} \sim Multi(NN_f(\boldsymbol{Z})), Y \sim Gaussian(\boldsymbol{w} \odot \boldsymbol{Z}, 1), \qquad (11)$$

where $Multi$ represents multinomial distribution, $NN_f$ is a neural network with softmax activation, $\boldsymbol{Z}, \boldsymbol{w} \in \mathbb{R}^{K_Z}$, $K_Z = 8$, and $\odot$ represents the element-wise product operator, respectively. We then treat the first $K_C = 5$ dimensions of $\boldsymbol{Z}$ as the latent confounders $\boldsymbol{C}$ and the remaining $K_M = K_Z - K_C$ dimensions as the latent mediators $\boldsymbol{M}$. After learning $NN_f$ and $\boldsymbol{w}$ according to Eq. (11), we draw latent confounders $\boldsymbol{C} \in Gaussian(0, \mathbf{I})$, latent mediators $\boldsymbol{M} = T \cdot \boldsymbol{\gamma}$, and set the outcome $Y = \boldsymbol{w} \odot [\boldsymbol{C} || \boldsymbol{M}] + \tau \cdot T$, where the true ATE can be calculated as $sum(\boldsymbol{\gamma} \odot \boldsymbol{w}_{-K_M:}) + \tau$.

### 5.1.1 Disentangle Confounders and Post-treatment Variables

We first show the $p$-value of the kernel-based pairwise independence test of the true latent variables before and after conditioning on the assigned treatment $T$. From Fig. 2, we can find that the distinction of the intra-confounder case from the other two cases discussed in Subsection 4.4 is significant. Here, we should note this relies on the assumption that latent confounders are independent. If the latent confounders are correlated, we can first use causal discovery techniques such as the PC algorithm [39] to find direct parents of $T$, and use our algorithm as the refinement to determine the true confounders $C$ from the misidentified post-treatment variables (Experiments see Section D) in Appendix.

## 5.2 Baselines

The baselines we include for comparisons can be categorized into three classes. *(i)* **Unawareness**, where no information in $\boldsymbol{X}$ is used for ATE estimation. We implement the naive LR0 estimator, which regresses $Y$ on $T$ and uses the coefficient to estimate the ATE [15] (LR0 equals to $DEV(\emptyset)$, i.e., the difference of the average outcome between the treatment and non-treatment group). *(ii)* **Control-$\boldsymbol{X}$**, which directly controls the covariates $\boldsymbol{X}$. In this class, LR1 regresses $Y$ on $T$ and $\boldsymbol{X}$, whereas TarNet uses a two-branch neural network to estimate the $DEV(\boldsymbol{X})$ *(iii)* **Control-$\boldsymbol{Z}$**, which controls latent variables $\boldsymbol{Z}$ learned from the covariates $\boldsymbol{X}$. Methods from this class include the CEVAE [25] and covariate disentanglement methods, such as DR-CFR [12], TEDVAE [44], NICE [38], and AFS [41].

### 5.2.1 Results and Analysis

From Table 1, we can find that for all four datasets, CEVAE is worse than the naive LR0 estimator. In addition, for the LatentMediator and Company (Age) dataset, all methods except CiVAE fail to predict the negativity of the ATE. Covariates disentanglement-based methods, i.e., DR-CFR and TEDVAE, inherit the latent post-treatment bias of CEVAE. The reason is that, these methods disentangle latent confounders $\boldsymbol{C}$ from latent instrumental variables $\boldsymbol{I}$ and latent adjusters $\boldsymbol{A}$ by

---

[3]which leads to 0.178 and -0.105 after standardization of the outcome.

Table 1: Comparison of CiVAE with baselines under latent post-treatment bias on various datasets.

| Dataset | LatentMediator | | LatentCorrelator | | Company (Age) | | Company (Gender) | |
|---|---|---|---|---|---|---|---|---|
| Method | ATE. | Err. | ATE. | Err. | ATE. | Err. | ATE. | Err. |
| LR0 | $0.975 \pm 0.032$ | 1.975 | $2.977 \pm 0.032$ | 1.977 | $0.131 \pm 0.015$ | 0.399 | $-0.105 \pm 0.009$ | -0.213 |
| LR1 | $1.457 \pm 0.167$ | 2.457 | $3.400 \pm 0.130$ | 2.400 | $0.093 \pm 0.029$ | 0.361 | $-0.175 \pm 0.014$ | -0.256 |
| TarNet | $1.461 \pm 0.172$ | 2.461 | $3.414 \pm 0.146$ | 2.414 | $0.112 \pm 0.085$ | 0.380 | $-0.167 \pm 0.021$ | -0.248 |
| CEVAE | $1.550 \pm 0.292$ | 2.550 | $3.323 \pm 0.167$ | 2.323 | $0.106 \pm 0.078$ | 0.374 | $-0.180 \pm 0.028$ | -0.261 |
| DR-CFR | $1.239 \pm 0.324$ | 2.239 | $3.185 \pm 0.319$ | 2.185 | $0.094 \pm 0.089$ | 0.362 | $-0.159 \pm 0.030$ | -0.240 |
| NICE | $1.868 \pm 0.530$ | 2.868 | $1.942 \pm 0.524$ | 0.942 | $0.149 \pm 0.126$ | 0.417 | $-0.186 \pm 0.041$ | -0.267 |
| TEDVAE | $1.042 \pm 0.315$ | 2.042 | $3.138 \pm 0.281$ | 2.138 | $0.097 \pm 0.093$ | 0.365 | $-0.143 \pm 0.027$ | -0.224 |
| AFS | $1.496 \pm 0.825$ | 2.496 | $3.251 \pm 0.398$ | 2.251 | $0.105 \pm 0.102$ | 0.373 | $-0.163 \pm 0.045$ | -0.244 |
| CiVAE | $\mathbf{-0.822} \pm 0.753$ | **0.178** | $\mathbf{1.199} \pm 0.765$ | **0.199** | $\mathbf{-0.140} \pm 0.137$ | **0.128** | $\mathbf{-0.106} \pm 0.064$ | **-0.187** |
| True ATE | $-1.000 \pm 0.000$ | 0.000 | $1.000 \pm 0.000$ | 0.000 | $-0.268 \pm 0.000$ | 0.000 | $-0.081 \pm 0.000$ | 0.000 |

utilizing their causal relations with $T$ and $Y$, i.e., $\boldsymbol{I}$ is predictive only for $T$, $\boldsymbol{A}$ is predictive only for $Y$, whereas $\boldsymbol{C}$ is predictive for both $T$ and $Y$. For example, TEDVAE includes three encoders to infer three sets of latent variables $\hat{\boldsymbol{I}}$, $\hat{\boldsymbol{A}}$, $\hat{\boldsymbol{C}}$ from $\boldsymbol{X}$ and adds classification losses $p(T|\hat{\boldsymbol{I}}, \hat{\boldsymbol{C}})$ and $p(Y|T, \hat{\boldsymbol{C}}, \hat{\boldsymbol{A}})$ on the CEVAE loss. However, since both latent confounders $\boldsymbol{C}$ and latent post-treatment variables $\boldsymbol{M}$ are correlated with both $T$ and $Y$, these methods cannot disentangle $\boldsymbol{C}$ from $\boldsymbol{M}$. An exception is NICE [38], which uses invariant risk minimization (IRM) [3] to find all causal parents of the outcome $Y$ as the confounders, which makes it more robust in the LatentCorrelator case. However, since mediators $\mathbf{M}$ are also the causal parent of $Y$, the performance degrades substantially on the LatentMediator dataset. Although AFS [41] considers the existence of post-treatment variables $\boldsymbol{M}$ in the proxy $\boldsymbol{X}$, it assumes that they can be separated from other variables in $\boldsymbol{X}$ in the observational space, and no relationship exists between the post-treatment variables and the outcome, so it still has poor performance in our setting since both assumptions are violated.

## 5.3 Sensitivity Analysis

In this part, we vary the number of confounders and post-treatment variables that generate proxy $\boldsymbol{X}$ in the Company (Age) and Company (Gender) datasets and compare CiVAE with the baseline TEDVAE in Fig. 3. Fig. 3 shows that the error is consistently lower for CiVAE. In addition, the error is comparatively higher when the number of confounders is low since the misidentification of latent post-treatment variables as confounders can have a com-

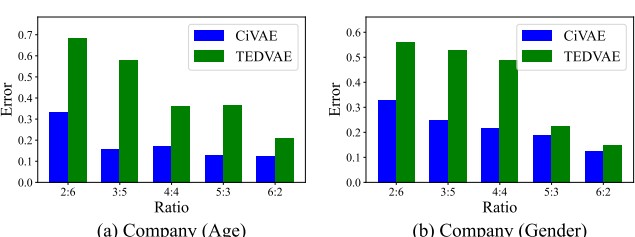

(a) Company (Age)   (b) Company (Gender)

Figure 3: Error with different ratio of latent confounders and latent post-treatment variable in the latent space.

paratively larger influence on the ATE estimation. In addition, when the number of confounders becomes larger, the performance gap between CiVAE and TEDVAE gracefully shrinks.

## 6 Conclusions

In this paper, we systematically investigate the latent post-treatment bias in causal inference from observational data. We first prove that unresolved latent post-treatment variables scrambled in the proxy of confounders can arbitrarily bias the ATE estimation. To address the bias, we proposed the Confounder-identifiable VAE (CiVAE), which, utilizing a mild assumption regarding the prior of latent factors, guarantees the identifiability of latent confounders up to bijective transformations. Finally, we show that controlling the latent confounders inferred by CiVAE can provide an unbiased estimation of the ATE. Experiments on both simulated and real-world datasets demonstrate that CiVAE has superior robustness to latent post-treatment bias compared to state-of-the-art methods.

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

# Appendix

## A  Broader Impact

The proposed CiVAE is a universal model for causal effect estimation with observational data. Although we use the Company job data that estimate the causal effects of *online working mode* to *applicant statistics* as a real-world example, proxy-of-confounder-based methods have been heavily used in other observational studies, which may be susceptible to latent post-treatment bias. Therefore, we speculate that the proposed CiVAE will have a broader impact on causal inference community.

## B  Related Work

### B.1  Post-Treatment Bias in Causal Inference

Bias due to accidentally controlling post-treatment variables, i.e., *post-treatment bias*, has long been recognized as dangerous in causal effect estimation [20]. Back at 2005, Pearl [32] cautioned that controlling more is not better, and uses the collider bias [9] and M-Bias [7] as two examples to show that bias can be increased when controlling the post-treatment variables. Furthermore, [30] show that indirect correlations between post-treatment variable $M$ and outcome $Y$ can still cause bias. Recent works prove that even if $M$ has no causal relationship with $Y$, controlling it can still increase the variance of estimand [12]. However, most of these works study the post-treatment bias in the observational space, where latent post-treatment variables that are mixed with confounders to generate the observed covariates can be easily ignored by the researcher. Therefore, it motivates us to develop CiVAE, which is robust to the latent post-treatment bias under mild assumptions.

### B.2  Covariate Disentanglement

Recently, researchers have realized that directly controlling proxy of confounders $\mathbf{X}$ may not be safe, as variables other than confounders could lurk in the proxy and ruin the ATE estimation [12]. Traditional methods assume that the variables that generate $\mathbf{X}$ are a mixture of confounders, adjusters, and influencers [36], where adjusters should not be controlled as it can increase the estimation variance [11]. Most methods rely on the fact that adjusters are correlated only with the treatment to separate them from other variables [12, 44] (see Fig. (1)). This can also be used to remove post-treatment variables that are not correlated with the outcome, which have similar statistics properties with adjustors [41]. Here, a different work is NICE [38], which uses the fact that confounders and influencers are direct causal parents of the outcome to find these variables with invariant learning as the control set [3]. However, since mediators are also direct parents of the outcome, NICE is still not robust to general post-treatment bias. Given that all above methods cannot satisfactorily address the latent post-treatment in general cases, it is imperative to design the CiVAE, where confounders can be identified and distinguished with latent post-treatment variables for unbiased adjustment.

## C  Theoretical Analysis

### C.1  Proof of Lemma 3.1.

*Proof.* Let $\mathbf{Z} = f(\mathbf{X})$ and $\mathbf{z} = f(\mathbf{x})$. If $f$ is injective and differentiable *a.e.*, and $f^{\dagger}$ is the left-inverse, we have:

$$f_{Y|f(\mathbf{X})}(y|f(\mathbf{x})) = f_{Y|\mathbf{Z}}(y|\mathbf{z}) = \frac{f_{Y,\mathbf{Z}}(y,\mathbf{z})}{f_{\mathbf{Z}}(\mathbf{z})} = \frac{f_{Y,\mathbf{X}}(y,f^{\dagger}(\mathbf{z}))|\mathbf{J}_{f^{\dagger}}(\mathbf{z})|}{f_{\mathbf{X}}(f^{\dagger}(\mathbf{z}))|\mathbf{J}_{f^{\dagger}}(\mathbf{z})|} = \frac{f_{Y,\mathbf{X}}(y,\mathbf{x})}{f_{\mathbf{X}}(\mathbf{x})} = f_{Y|\mathbf{X}}(y|\mathbf{x}),$$

(12)

where $f_{\cdot}$ and $f_{\cdot|\cdot}$ represent the marginal and conditional density function, respectively, and $\mathbf{J}_{f^{\dagger}}(\mathbf{z})$ is the Jacobian matrix of function $f^{\dagger}$ evaluated at $\mathbf{z}$. Based on Eq. (12), we have:

$$\mathbb{E}[Y|\mathbf{X}] = \int \mathbf{y} \cdot f_{Y|\mathbf{X}}(\mathbf{y}|\mathbf{x})dy = \int y \cdot f_{Y|\mathbf{Z}}(\mathbf{y}|\mathbf{z})dy = \mathbb{E}[Y|\mathbf{Z} = \mathbf{z}] = \mathbb{E}[Y|f(\mathbf{X}) = f(\mathbf{x})]. \quad (13)$$

$\square$

## C.2 Proof of Corollary 3.3.

*Proof.* For $\boldsymbol{X} = \boldsymbol{x}$, let $[\boldsymbol{c}||\boldsymbol{m}] \doteq [f_C^\dagger(\boldsymbol{x})||f_M^\dagger(\boldsymbol{x})] \doteq f^\dagger(\boldsymbol{x}) = \mathbf{A}^\dagger(\boldsymbol{x} - \boldsymbol{\alpha}_X)$, where $\mathbf{A}^\dagger$ is the left inverse of the full column-rank matrix $\mathbf{A}$ in Eq. (2), we have:

$$
\begin{aligned}
CATE(\boldsymbol{x}) &= \mathbb{E}[Y|T = 1, \boldsymbol{C} = f_C^\dagger(\boldsymbol{x})] - \mathbb{E}[Y|T = 0, \boldsymbol{C} = f_C^\dagger(\boldsymbol{x})] \\
&= \mathbb{E}[Y|T = 1, \boldsymbol{C} = \boldsymbol{c}] - \mathbb{E}[Y|T = 0, \boldsymbol{C} = \boldsymbol{c}] \\
&= \mathbb{E}[\alpha_Y + \tau \cdot T + \sum \theta_j \cdot M_j + \sum \kappa_i \cdot C_i | T = 1, \boldsymbol{C} = \boldsymbol{c}] \\
&\quad - \mathbb{E}[\alpha_Y + \tau \cdot T + \sum \theta_j \cdot M_j + \sum \kappa_i \cdot C_i | T = 0, \boldsymbol{C} = \boldsymbol{c}] \\
&= \alpha_Y + \tau \cdot \mathbb{E}[T|T = 1, \boldsymbol{C} = \boldsymbol{c}] + \sum \theta_j \cdot \mathbb{E}[M_j|T = 1, \boldsymbol{C} = \boldsymbol{c}] + \sum \kappa_i \cdot \mathbb{E}[C_i|T = 1, \boldsymbol{C} = \boldsymbol{c}] \\
&\quad - \alpha_Y + \tau \cdot \mathbb{E}[T|T = 0, \boldsymbol{C} = \boldsymbol{c}] + \sum \theta_j \cdot \mathbb{E}[M_j|T = 0, \boldsymbol{C} = \boldsymbol{c}] + \sum \kappa_i \cdot \mathbb{E}[C_i|T = 0, \boldsymbol{C} = \boldsymbol{c}] \\
&= \tau \cdot (1 - 0) + \sum \theta_j \cdot (\gamma_j \cdot (1 - 0)) + \sum \kappa_i \cdot (c_i - c_i) \\
&= \tau + \sum \theta_j \cdot \gamma_j = \mathbb{E}[\tau + \sum \theta_j \cdot \gamma_j] = ATE,
\end{aligned}
\tag{14}
$$

where the first equality is due to the definition of CATE in Eq. (2). In addition, the causal estimand and bias of a proxy-of-confounder-based causal inference model that controls the latent variable $\boldsymbol{Z}$ inferred via $\boldsymbol{Z} = \bar{f}(\boldsymbol{X}) = \mathbf{B}^T \boldsymbol{X}$ (where $\mathbf{B}$ is also a full column-rank matrix) can be formulated as:

$$
\begin{aligned}
DCEV(\mathbf{B}^T \boldsymbol{x}) &= \mathbb{E}[Y|T = 1, \boldsymbol{Z} = \mathbf{B}^T \boldsymbol{x}] - \mathbb{E}[Y|T = 0, \boldsymbol{Z} = \mathbf{B}^T \boldsymbol{x}] \\
&= \mathbb{E}[Y|T = 1, \boldsymbol{Z} = \mathbf{B}^T \boldsymbol{\alpha}_X + \mathbf{B}^T \mathbf{A}[\boldsymbol{c}||\boldsymbol{m}]] - \mathbb{E}[Y|T = 0, \boldsymbol{Z} = \mathbf{B}^T \boldsymbol{\alpha}_X + \mathbf{B}^T \mathbf{A}[\boldsymbol{c}||\boldsymbol{m}]] \\
&\overset{(a)}{=} \mathbb{E}[Y|T = 1, \boldsymbol{C} = \boldsymbol{c}, \boldsymbol{M} = \boldsymbol{m}] - \mathbb{E}[Y|T = 0, \boldsymbol{C} = \boldsymbol{c}, \boldsymbol{M} = \boldsymbol{m}] \\
&= \alpha_Y + \tau \cdot 1 + \sum \theta_j \cdot \mathbb{E}[M_j|T = 1, \boldsymbol{C} = \boldsymbol{c}, \boldsymbol{M} = \boldsymbol{m}] + \sum \kappa_i \cdot \mathbb{E}[C_i|T = 1, \boldsymbol{C} = \boldsymbol{c}, \boldsymbol{M} = \boldsymbol{m}] \\
&\quad - \alpha_Y + \tau \cdot 0 + \sum \theta_j \cdot \mathbb{E}[M_j|T = 0, \boldsymbol{C} = \boldsymbol{c}, \boldsymbol{M} = \boldsymbol{m}] + \sum \kappa_i \cdot \mathbb{E}[C_i|T = 0, \boldsymbol{C} = \boldsymbol{c}, \boldsymbol{M} = \boldsymbol{m}] \\
&= \tau \cdot (1 - 0) + \sum \theta_j \cdot (m_j - m_j) + \sum \kappa_i \cdot (c_i - c_i) \\
&= \tau = \mathbb{E}[\tau] = \mathbb{E}[DCEV(\mathbf{B}^T \boldsymbol{X})],
\end{aligned}
\tag{15}
$$

where step (a) is due to the fact that, since both $\mathbf{A}$ and $\mathbf{B}$ are full column-rank matrices, $\mathbf{B}^T \mathbf{A}$ is an invertible matrix, and the mapping $f = \mathbf{B}^T \boldsymbol{\alpha}_X + \mathbf{B}^T \mathbf{A}$ is bijective. Therefore, we can invoke Lemma 3.1 and apply the left-inverse of $f$, i.e., $f^\dagger = (\mathbf{B}^T \mathbf{A})^{-1} - \mathbf{B}^T \boldsymbol{\alpha}_X$, to the condition of the expectation. The rest steps are based on the structural causal equations defined in Eq. (2). $\square$

## C.3 Another Case of Linear SCM with Latent Correlators

**Corollary C.1.** *For another Linear Structural Causal Model defined as follows*

$$
\begin{aligned}
T &\leftarrow \mathbb{1}(\alpha_T + \sum \beta_i \cdot C_i > a) \\
M_j &\leftarrow \alpha_M + \gamma_j \cdot T + {\color{red}\phi_j \cdot U_j} \\
\boldsymbol{X} &\leftarrow \boldsymbol{\alpha}_X + \mathbf{A}[\boldsymbol{M}||\boldsymbol{C}] \\
Y &\leftarrow \alpha_Y + \tau \cdot T + {\color{red}\sum \theta_j \cdot U_j} + \sum \kappa_i \cdot C_i,
\end{aligned}
\tag{16}
$$

*where $f = \mathbf{A} \in \mathbb{R}^{K_X \times (K_C + K_M)}$ is a full column-rank matrix, the CATE, ATE, and the bias of proxy-of-confounder-based causal inference model that controls the latent variable $\boldsymbol{Z}$ inferred via $\boldsymbol{Z} = \bar{f}(\boldsymbol{X}) = \mathbf{B}^T \boldsymbol{X}$ can be formulated as follows:*

$$
ATE = CATE = {\color{red}\tau}
$$

$$
\mathbb{E}[DCEV(\boldsymbol{Z} = \mathbf{B}^T \boldsymbol{X})] = DCEV(\boldsymbol{Z} = \mathbf{B}^T \boldsymbol{X}) = \tau - {\color{red}\sum \frac{\theta_j \cdot \gamma_j}{\phi_j}}
$$

$$
Bias = ATE - \mathbb{E}[DCEV(\mathbf{B}^T \boldsymbol{X})] = \sum \frac{\theta_j \cdot \gamma_j}{\phi_j},
\tag{17}
$$

530 *where* $\mathbf{B} \in \mathbb{R}^{K_X \times (K_C + K_M)}$ *is another full column-rank matrix. Since* $\sum \frac{\theta_j \cdot \gamma_j}{\phi_j}$ *is arbitrary, the*
531 *estimator* $\mathbb{E}[DCEV(\mathbf{B}^T \boldsymbol{X})]$ *is arbitrarily biased for the estimation of ATE.*

532 *Proof.* The proof of the CATE and ATE is trivial. The causal estimand and the bias of a proxy-
533 of-confounder-based causal inference model that controls the latent variables $\boldsymbol{Z}$ inferred via $\boldsymbol{Z} =$
534 $\bar{f}(\boldsymbol{X}) = \mathbf{B}^T \boldsymbol{X}$ (where $\mathbf{B}$ is also a full column-rank matrix) can be formulated as follows:

$$
\begin{aligned}
DCEV(\mathbf{B}^T \boldsymbol{x}) &= \mathbb{E}[Y|T = 1, \boldsymbol{Z} = \mathbf{B}^T \boldsymbol{x}] - \mathbb{E}[Y|T = 0, \boldsymbol{Z} = \mathbf{B}^T \boldsymbol{x}] \\
&= \mathbb{E}[Y|T = 1, \boldsymbol{Z} = \boldsymbol{\alpha}_X + \mathbf{B}^T \mathbf{A}[\boldsymbol{c}||\boldsymbol{m}]] - \mathbb{E}[Y|T = 0, \boldsymbol{Z} = \boldsymbol{\alpha}_X + \mathbf{B}^T \mathbf{A}[\boldsymbol{c}||\boldsymbol{m}]] \\
&\overset{(a)}{=} \mathbb{E}[Y|T = 1, \boldsymbol{C} = \boldsymbol{c}, \boldsymbol{M} = \boldsymbol{m}] - \mathbb{E}[Y|T = 0, \boldsymbol{C} = \boldsymbol{c}, \boldsymbol{M} = \boldsymbol{m}] \\
&= \alpha_Y + \tau \cdot 1 + \sum \theta_j \cdot \mathbb{E}[U_j|T = 1, \boldsymbol{C} = \boldsymbol{c}, \boldsymbol{M} = \boldsymbol{m}] + \sum \kappa_i \cdot \mathbb{E}[C_i|T = 1, \boldsymbol{C} = \boldsymbol{c}, \boldsymbol{M} = \boldsymbol{m}] \\
&\quad - \alpha_Y + \tau \cdot 0 + \sum \theta_j \cdot \mathbb{E}[U_j|T = 0, \boldsymbol{C} = \boldsymbol{c}, \boldsymbol{M} = \boldsymbol{m}] + \sum \kappa_i \cdot \mathbb{E}[C_i|T = 0, \boldsymbol{C} = \boldsymbol{c}, \boldsymbol{M} = \boldsymbol{m}] \\
&= \tau \cdot (1 - 0) + \sum \theta_j \cdot (\phi_j^{-1} \cdot (m_j - \alpha_M - \gamma_j) - \phi_j^{-1} \cdot (m_j - \alpha_M)) + \sum \kappa_i \cdot (c_i - c_i) \\
&= \tau - \sum \frac{\theta_j \cdot \gamma_j}{\phi_j} = \mathbb{E}\left[\tau - \sum \frac{\theta_j \cdot \gamma_j}{\phi_j}\right] = \mathbb{E}[DCEV(\mathbf{B}^T \boldsymbol{X})],
\end{aligned}
$$

(18)

535 □

536 where step (a) and the rest of the proof follow the same logic as the proof in Section 3.3.

537 ## C.4 Proof of Theorem 4.1

538 The strict definitions of the exponential family, strong exponential (which is assumed for the factorized
539 part of the conditional prior), and identifiability follow [19, 26], and can be referred to in Appendix
540 E, F of [26], which we omit to avoid redundancy. The proof of Theorem 4.1 is largely based on the
541 NF-iVAE paper [26], where most of the details can be found, with the new assumption introduced in
542 CiVAE that each $\boldsymbol{S}_{f,i}$ has at least one invertible dimension incorporated to ensure that each dimension
543 of the inferred latent variables is a bijective transformation of the corresponding true latent variable.

544 ### C.4.1 PART I

545 **Step I**. In this step, we transform the equality of noisy conditional marginal distribution of $\boldsymbol{X}$ given
546 $Y, T$ of two models with parameter $\boldsymbol{\theta}, \tilde{\boldsymbol{\theta}} \in \Theta$ into the equality of noise-free distributions.

$$
\begin{aligned}
&p_{\boldsymbol{\theta}}(\boldsymbol{X} \mid Y, T) = p_{\tilde{\boldsymbol{\theta}}}(\boldsymbol{X} \mid Y, T) \\
&\implies \int_{\mathcal{Z}} p_f(\boldsymbol{X} \mid \boldsymbol{Z}) p_{\boldsymbol{S}, \boldsymbol{\lambda}}(\boldsymbol{Z} \mid Y, T) d\boldsymbol{Z} = \int_{\mathcal{Z}} p_{\tilde{f}}(\boldsymbol{X} \mid \boldsymbol{Z}) p_{\tilde{\boldsymbol{S}}, \tilde{\boldsymbol{\lambda}}}(\boldsymbol{Z} \mid Y, T) d\boldsymbol{Z} \\
&\implies \int_{\mathcal{Z}} p_{\boldsymbol{\varepsilon}}(\boldsymbol{X} - f(\boldsymbol{Z})) p_{\boldsymbol{S}, \boldsymbol{\lambda}}(\boldsymbol{Z} \mid Y, T) d\boldsymbol{Z} = \int_{\mathcal{Z}} p_{\boldsymbol{\varepsilon}}(\boldsymbol{X} - \tilde{f}(\boldsymbol{Z})) p_{\tilde{\boldsymbol{S}}, \tilde{\boldsymbol{\lambda}}}(\boldsymbol{Z} \mid Y, T) d\boldsymbol{Z} \\
&\overset{(a)}{\implies} \int_{\mathcal{X}} p_{\boldsymbol{\varepsilon}}(\boldsymbol{X} - \overline{\boldsymbol{X}}) p_{\boldsymbol{S}, \boldsymbol{\lambda}}\left(f^\dagger(\overline{\boldsymbol{X}}) \mid Y, T\right) \mathrm{vol}\left(\mathbf{J}_{f^\dagger}(\overline{\boldsymbol{X}})\right) d\overline{\boldsymbol{X}} = \\
&\qquad \int_{\mathcal{X}} p_{\boldsymbol{\varepsilon}}(\boldsymbol{X} - \overline{\boldsymbol{X}}) p_{\tilde{\boldsymbol{S}}, \tilde{\boldsymbol{\lambda}}}\left(\tilde{f}^\dagger(\overline{\boldsymbol{X}}) \mid Y, T\right) \mathrm{vol}\left(\mathbf{J}_{\tilde{f}^\dagger}(\overline{\boldsymbol{X}})\right) d\overline{\boldsymbol{X}} \\
&\overset{(b)}{\implies} \int_{\mathbb{R}^d} p_{\boldsymbol{\varepsilon}}(\boldsymbol{X} - \overline{\boldsymbol{X}}) \tilde{p}_{f, \boldsymbol{S}, \boldsymbol{\lambda}, Y, T}(\overline{\boldsymbol{X}}) d\overline{\boldsymbol{X}} = \int_{\mathbb{R}^d} p_{\boldsymbol{\varepsilon}}(\boldsymbol{X} - \overline{\boldsymbol{X}}) \tilde{p}_{\tilde{f}, \tilde{\boldsymbol{S}}, \tilde{\boldsymbol{\lambda}}, \tilde{Y}, \tilde{T}}(\overline{\boldsymbol{X}}) d\overline{\boldsymbol{X}} \\
&\implies \left(\tilde{p}_{f, \boldsymbol{S}, \boldsymbol{\lambda}, Y, T} * p_{\boldsymbol{\varepsilon}}\right)(\boldsymbol{X}) = \left(\tilde{p}_{\tilde{f}, \tilde{\boldsymbol{S}}, \tilde{\boldsymbol{\lambda}}, \tilde{Y}, \tilde{T}} * p_{\boldsymbol{\varepsilon}}\right)(\boldsymbol{X}) \\
&\overset{(c)}{\implies} F\left[\tilde{p}_{f, \boldsymbol{S}, \boldsymbol{\lambda}, Y, T}\right](\boldsymbol{\omega}) \varphi_{\boldsymbol{\varepsilon}}(\boldsymbol{\omega}) = F\left[\tilde{p}_{\tilde{f}, \tilde{\boldsymbol{S}}, \tilde{\boldsymbol{\lambda}}, \tilde{Y}, \tilde{T}}\right](\boldsymbol{\omega}) \varphi_{\boldsymbol{\varepsilon}}(\boldsymbol{\omega}) \\
&\overset{(d)}{\implies} F\left[\tilde{p}_{f, \boldsymbol{S}, \boldsymbol{\lambda}, Y, T}\right](\boldsymbol{\omega}) = F\left[\tilde{p}_{\tilde{f}, \tilde{\boldsymbol{S}}, \tilde{\boldsymbol{\lambda}}, \tilde{Y}, \tilde{T}}\right](\boldsymbol{\omega}) \\
&\implies \tilde{p}_{f, \boldsymbol{S}, \boldsymbol{\lambda}, Y, T}(\boldsymbol{X}) = \tilde{p}_{\tilde{f}, \tilde{\boldsymbol{S}}, \tilde{\boldsymbol{\lambda}}, \tilde{Y}, \tilde{T}}(\boldsymbol{X}).
\end{aligned}
$$

(19)

547 Step (a) is based on the rule of change-of-variable, where $\text{vol}(\mathbf{A}) = \sqrt{\det\left(\mathbf{A}^T\mathbf{A}\right)}$. In step (b),

548 we define $\tilde{p}_{f,\boldsymbol{S},\boldsymbol{\lambda},Y,T}(\boldsymbol{X}) \triangleq p_{\boldsymbol{S},\boldsymbol{\lambda}}\left(f^\dagger(\boldsymbol{X}) \mid Y, T\right) \text{vol}\left(\mathbf{J}_{f^\dagger}(\boldsymbol{X})\right)\mathbb{I}_\mathcal{X}(\boldsymbol{X})$. In step (c), we use $F[\cdot]$ to

549 denote the Fourier transform. In step (d), we drop $\varphi_\varepsilon(\boldsymbol{\omega})$ as it is non-zero *a.e.* (see Assumption 3).

550 **Step II**. In this step, we transform the equality of the noise-free distributions into the relationship of

551 the sufficient statistics $\boldsymbol{S}$ and $\tilde{\boldsymbol{S}}$. By taking logarithm of both sides of Eq. (19), we have:

$$\log \text{vol}\left(J_{f^\dagger}(\boldsymbol{X})\right) + \log \mathcal{Q}\left(f^\dagger(\boldsymbol{X})\right) - \log \mathcal{C}(Y, T) + \left\langle \boldsymbol{S}\left(f^\dagger(\boldsymbol{X})\right), \boldsymbol{\lambda}(Y, T)\right\rangle$$
$$= \log \text{vol}\left(J_{\tilde{f}^\dagger}(\boldsymbol{X})\right) + \log \tilde{\mathcal{Q}}\left(\tilde{f}^\dagger(\boldsymbol{X})\right) - \log \tilde{\mathcal{C}}(Y, T) + \left\langle \tilde{\boldsymbol{S}}\left(\tilde{f}^\dagger(\boldsymbol{X})\right), \tilde{\boldsymbol{\lambda}}(Y, T)\right\rangle. \quad (20)$$

552 Let $(Y,T)_0, \cdots, (Y,T)_k$ be the $k+1$ distinct points defined in Assumption 3 - (iv). We obtain $k+1$

553 equations by evaluating the Eq. (20) at these points, where the first equation is subtracted from the

554 remaining ones, which leads to the following equation system:

$$\left\langle \boldsymbol{S}\left(f^\dagger(\boldsymbol{X})\right), \boldsymbol{\lambda}\left((Y,T)_l\right) - \boldsymbol{\lambda}\left((Y,T)_0\right)\right\rangle + \log \frac{\mathcal{C}\left((Y,T)_0\right)}{\mathcal{C}\left((Y,T)_l\right)}$$
$$= \left\langle \tilde{\boldsymbol{S}}\left(\tilde{f}^\dagger(\boldsymbol{X})\right), \tilde{\boldsymbol{\lambda}}\left((Y,T)_l\right) - \tilde{\boldsymbol{\lambda}}\left((Y,T)_0\right)\right\rangle + \log \frac{\tilde{\mathcal{C}}\left((Y,T)_0\right)}{\tilde{\mathcal{C}}\left((Y,T)_l\right)}, \quad l = 1, \cdots, k. \quad (21)$$

555 Let $\mathbf{L}$ be the invertible matrix defined in Assumption 3 - (iv) and $\tilde{\mathbf{L}}$ be the counterpart for $\tilde{\boldsymbol{\lambda}}$, if we

556 summarize all terms irrelevant to $\boldsymbol{X}$ into a constant $\boldsymbol{b}$, we have:

$$\mathbf{L}^T \boldsymbol{S}\left(f^\dagger(\boldsymbol{X})\right) = \tilde{\mathbf{L}}^T \tilde{\boldsymbol{S}}\left(\tilde{f}^\dagger(\boldsymbol{X})\right) + \boldsymbol{b}$$
$$\implies \boldsymbol{S}\left(f^\dagger(\boldsymbol{X})\right) = \mathbf{A}\tilde{\boldsymbol{S}}\left(\tilde{f}^\dagger(\boldsymbol{X})\right) + \boldsymbol{c}, \quad (22)$$

557 where $\mathbf{A} = \mathbf{L}^{-T}\tilde{\mathbf{L}} \in \mathbb{R}^{k \times k}$, and $\boldsymbol{c} = \mathbf{L}^{-T}\boldsymbol{b} \in \mathbb{R}^k$.

558 **Step III**. Ideally, to prove the element-wise bijective identifiability of the latent variables $\boldsymbol{Z}$, the

559 transformation of the sufficient statistics $\boldsymbol{S}$ derived in Eq. (22) should be bijective. We claim that if

560 the conditional prior $p_{\boldsymbol{S},\boldsymbol{\lambda}}(\boldsymbol{Z} \mid Y, T)$ is strongly exponential and $\mathbf{L}$ is invertible, $\tilde{\mathbf{L}}$ and $\mathbf{A}$ must also

561 be invertible. The proof is omitted, and can be referred to in Appendix H.1.1 of [26].

## C.4.2 PART II

563 In this part, we prove that, if Assumptions 1, 2 and 3 hold, we can identify the factorized part

564 of the sufficient statistics $\boldsymbol{S}(\boldsymbol{Z})$, i.e., $\boldsymbol{S}_f(\boldsymbol{Z})$, up to permutation and element-wise transformation.

565 Specifically, if we use $\boldsymbol{v}$ to denote the composite map $\tilde{f}^\dagger \circ f : \mathcal{Z} \to \mathcal{Z}$, Eq. (22) can be rewritten into:

$$\boldsymbol{S}(\boldsymbol{Z}) = \mathbf{A}\tilde{\boldsymbol{S}}(\boldsymbol{v}(\boldsymbol{Z})) + \boldsymbol{c}. \quad (23)$$

566 We aim to prove that $\mathbf{A}$ in Eq. (23) is a block permutation matrix.

567 **Step I**. We start by showing that $\boldsymbol{v}$ is a component-wise function. If we differentiate both sides of Eq.

568 (23) with respect to $Z_s$ and $Z_t$, where $s \neq t$, we have:

$$\frac{\partial \boldsymbol{S}(\boldsymbol{Z})}{\partial Z_s} = \mathbf{A}\sum_{i=1}^{K_Z} \frac{\partial \tilde{\boldsymbol{S}}(\boldsymbol{v}(\boldsymbol{Z}))}{\partial v_i(\boldsymbol{Z})} \cdot \frac{\partial v_i(\boldsymbol{Z})}{\partial Z_s}$$
$$\frac{\partial^2 \boldsymbol{S}(\boldsymbol{Z})}{\partial Z_s \partial Z_t} = \mathbf{A}\sum_{i=1}^{K_Z}\sum_{i=1}^{K_Z} \frac{\partial^2 \tilde{\boldsymbol{S}}(\boldsymbol{v}(\boldsymbol{Z}))}{\partial v_i(\boldsymbol{Z})\partial v_j(\boldsymbol{Z})} \cdot \frac{\partial v_j(\boldsymbol{Z})}{\partial Z_t} \cdot \frac{\partial v_i(\boldsymbol{Z})}{\partial Z_s} + \mathbf{A}\sum_{i=1}^{K_Z} \frac{\partial \tilde{\boldsymbol{S}}(\boldsymbol{v}(\boldsymbol{Z}))}{\partial v_i(\boldsymbol{Z})} \cdot \frac{\partial^2 v_i(\boldsymbol{Z})}{\partial Z_s \partial Z_t}. \quad (24)$$

569 Note that for the factorized part of the sufficient statistics $\boldsymbol{S}$, i.e., $\boldsymbol{S}_f$, all *cross-derivatives* are zero,

570 and for the non-factorized part of $\boldsymbol{S}$, i.e., $\boldsymbol{S}_{nf}$, which is a neural network with ReLU activation (i.e.,

571 linear *a.e.*), all *second-order derivatives* are zero. Therefore, the *second order cross-derivatives* on

572 the LHS. of Eq. (24) are zero, which leads to the following equality:

$$\mathbf{0} = \mathbf{A}\sum_{i=1}^{K_Z} \frac{\partial^2 \tilde{\boldsymbol{S}}(\boldsymbol{v}(\boldsymbol{Z}))}{\partial v_i(\boldsymbol{Z})^2} \cdot \frac{\partial v_i(\boldsymbol{Z})}{\partial Z_t} \cdot \frac{\partial v_i(\boldsymbol{Z})}{\partial Z_s} + \mathbf{A}\sum_{i=1}^{K_Z} \frac{\partial \tilde{\boldsymbol{S}}(\boldsymbol{v}(\boldsymbol{Z}))}{\partial v_i(\boldsymbol{Z})} \cdot \frac{\partial^2 v_i(\boldsymbol{Z})}{\partial Z_s \partial Z_t}. \quad (25)$$

Eq. (25) can be written into the matrix-vector product form as follows:

$$0 = \mathbf{A}\tilde{\boldsymbol{S}}''(\boldsymbol{Z})\boldsymbol{v}'_{s,t}(\boldsymbol{Z}) + \mathbf{A}\tilde{\boldsymbol{S}}'(\boldsymbol{Z})\boldsymbol{v}''_{s,t}(\boldsymbol{Z}), \tag{26}$$

where

$$\tilde{\boldsymbol{S}}''(\boldsymbol{Z}) = \left[\frac{\partial^2 \tilde{\boldsymbol{S}}(\boldsymbol{v}(\boldsymbol{Z}))}{\partial v_1(\boldsymbol{Z})^2}, \cdots, \frac{\partial^2 \tilde{\boldsymbol{S}}(\boldsymbol{v}(\boldsymbol{Z}))}{\partial v_{K_Z}(\boldsymbol{Z})^2}\right] \in \mathbb{R}^{k \times K_Z},$$

$$\boldsymbol{v}'_{s,t}(\boldsymbol{Z}) = \left[\frac{\partial v_1(\boldsymbol{Z})}{\partial Z_t} \cdot \frac{\partial v_1(\boldsymbol{Z})}{\partial Z_s}, \cdots, \frac{\partial v_{K_Z}(\boldsymbol{Z})}{\partial Z_t} \cdot \frac{\partial v_{K_Z}(\boldsymbol{Z})}{\partial Z_s}\right]^T \in \mathbb{R}^{K_Z},$$

and

$$\tilde{\boldsymbol{S}}'(\boldsymbol{Z}) = \left[\frac{\partial \tilde{\boldsymbol{S}}(\boldsymbol{v}(\boldsymbol{Z}))}{\partial v_1(\boldsymbol{Z})}, \cdots, \frac{\partial \tilde{\boldsymbol{S}}(\boldsymbol{v}(\boldsymbol{Z}))}{\partial v_{K_Z}(\boldsymbol{Z})}\right] \in \mathbb{R}^{k \times K_Z},$$

$$\boldsymbol{v}''_{s,t}(\boldsymbol{Z}) = \left[\frac{\partial^2 v_1(\boldsymbol{Z})}{\partial Z_s \partial Z_t}, \cdots, \frac{\partial^2 v_{K_Z}(\boldsymbol{Z})}{\partial Z_s \partial Z_t}\right]^T \in \mathbb{R}^{K_Z}.$$

If we denote the concatenation as $\tilde{\boldsymbol{S}}'''(\boldsymbol{Z}) = \left[\tilde{\boldsymbol{S}}''(\boldsymbol{Z}), \tilde{\boldsymbol{S}}'(\boldsymbol{Z})\right] \in \mathbb{R}^{k \times 2K_Z}$ and $\boldsymbol{v}'''_{s,t}(\boldsymbol{Z}) = \left[\boldsymbol{v}'_{s,t}(\boldsymbol{Z})^T, \boldsymbol{v}''_{s,t}(\boldsymbol{Z})^T\right]^T \in \mathbb{R}^{2K_z}$, we have:

$$0 = \mathbf{A}\tilde{\boldsymbol{S}}'''(\boldsymbol{Z})\boldsymbol{v}'''_{s,t}(\boldsymbol{Z}). \tag{27}$$

Finally, if we denote the rows of $\tilde{\boldsymbol{S}}'''(\boldsymbol{Z})$ that correspond to the factorized part of $\boldsymbol{S}$ by $\tilde{\boldsymbol{S}}'''_f(\boldsymbol{Z})$, according to Lemma 5 of the iVAE paper [19] and the assumption that $k \geq 2K_Z$, we have that the rank of $\tilde{\boldsymbol{S}}'''_f(\boldsymbol{Z})$ is $2K_Z$. Since $k \geq 2K_Z$, the rank of $\tilde{\boldsymbol{S}}'''_f(\boldsymbol{Z})$ is also $2K_Z$. Since the rank of $\mathbf{A}$ is $k$, the rank of $\mathbf{A}\tilde{\boldsymbol{S}}'''(\boldsymbol{Z})$ is $2K_Z$, which implies that $\boldsymbol{v}'''_{s,t}(\boldsymbol{Z}) \in \mathbb{R}^{2K_Z}$ is a zero vector. Therefore, we have $\boldsymbol{v}'_{s,t}(\boldsymbol{Z}) = \mathbf{0}, \forall s \neq t$, and we have demonstrated that $\boldsymbol{v}$ is a component-wise function.

**Step II**. Based on **Step I**, we demonstrate that $\mathbf{A}$ is a block permutation matrix. Without loss of generality, we assume that the permutation in $\boldsymbol{v}$ is Identity, where $\boldsymbol{v}(\boldsymbol{Z}) = [v_1(Z_1), \cdots, v_{K_Z}(Z_{K_Z})]^T$ and each $v_i$ is a nonlinear univariate scalar function. Since $f$ and $\tilde{f}$ are injective, $\boldsymbol{v}$ is bijective and $\boldsymbol{v}^{-1}(\boldsymbol{Z}) = \left[v_1^{-1}(Z_1), \cdots, v_{K_Z}^{-1}(Z_{K_Z})\right]^T$. If we denote $\overline{\boldsymbol{S}}(\boldsymbol{v}(\boldsymbol{Z})) = \tilde{\boldsymbol{S}}(\boldsymbol{v}(\boldsymbol{Z})) + \mathbf{A}^{-1}\boldsymbol{c}$, Eq. (23) can be reformulated as $\boldsymbol{S}(\boldsymbol{Z}) = \mathbf{A}\overline{\boldsymbol{S}}(\boldsymbol{v}(\boldsymbol{Z}))$. We then apply $\boldsymbol{v}^{-1}$ to $\boldsymbol{Z}$ on both sides, which gives

$$\boldsymbol{S}\left(\boldsymbol{v}^{-1}(\boldsymbol{Z})\right) = \mathbf{A}\overline{\boldsymbol{S}}(\boldsymbol{Z}). \tag{28}$$

Let $t$ be the index of an entry in $\boldsymbol{S}$ that corresponds to the factorized part $\boldsymbol{S}_f$. For all $s \neq t$, we have:

$$0 = \frac{\partial \boldsymbol{S}\left(\boldsymbol{v}^{-1}(\boldsymbol{Z})\right)_t}{\partial Z_s} = \sum_{j=1}^{k} a_{tj} \frac{\partial \overline{\boldsymbol{S}}(\boldsymbol{Z})_j}{\partial Z_s}. \tag{29}$$

Since the entries of $\tilde{\boldsymbol{S}}$ are linearly independent, $a_{tj}$ is zero for any $j$ such that $\frac{\partial \overline{\boldsymbol{S}}(\boldsymbol{Z})_j}{\partial Z_s} \neq 0$. This includes the entries $S_j$ that correspond to (1) the factorized part that does not depend on $Z_t$; and (2) the non-factorized part $\boldsymbol{S}_{nf}$. Therefore, when $t$ is the index of an entry in the sufficient statistics $\boldsymbol{S}$ that corresponds to factor $i$ in the factorized part $\boldsymbol{S}_f$, i.e., $\boldsymbol{S}_{f,i}$, the only non-zero $a_{tj}$ are the ones that map between $\boldsymbol{S}_{f,i}(Z_i)$ and $\overline{\boldsymbol{S}}_{f,i}(v_i(Z_i))$. Therefore, we can construct an invertible submatrix $\mathbf{A}'_i$ with all non-zero elements $a_{tj}$ for all $t$ that corresponds to factor $i$, such that

$$\boldsymbol{S}_{f,i}(Z_i) = \mathbf{A}'_i \overline{\boldsymbol{S}}_{f,i}(v_i(Z_i)) = \mathbf{A}'_i \tilde{\boldsymbol{S}}_{f,i}(v_i(Z_i)) + \boldsymbol{c}_i, \quad i = 1, \cdots, K_Z, \tag{30}$$

where $\boldsymbol{c}_i$ denotes the corresponding elements of $\boldsymbol{c}$. Eq. (30) means that for each $i = 1, \cdots, K_Z$, the matrix block $\mathbf{A}'_i$ of $\mathbf{A}$ affinely transforms the $i$-specific sufficient statistics vector $\boldsymbol{S}_{f,i}(Z_i)$ into $\tilde{\boldsymbol{S}}_{f,i}(v_i(Z_i))$. In addition, there is also an additional block $\mathbf{A}'$ that affinely transforms $\boldsymbol{S}_{nf}(\boldsymbol{Z})$ in into $\boldsymbol{S}_{nf}(v(\boldsymbol{Z}))$. This completes the proof that $\mathbf{A}$ is a block permutation matrix.

 **C.4.3  PART III**

Let $\tilde{Z}_i = v_i\,(Z_i) = \tilde{f}^\dagger(\boldsymbol{X})_i$ be the $i$th inferred latent variable. Assume again that the permutation in $\boldsymbol{v}$ is Identity. In this part, we prove that if Assumption 2 holds, each inferred latent variable $\tilde{Z}_i$ is the bijective transformation of the true latent variable. The proof is as follows.

*Proof.* Plugging $\tilde{Z}_i$ into Eq. (30), we have:

$$\boldsymbol{S}_{f,i}(Z_i) = \mathbf{A}'_i \bar{\boldsymbol{S}}_{f,i}(\tilde{Z}_i). \tag{31}$$

According to Assumption 2, there exists one dimension of $\boldsymbol{S}_{f,i}$, i.e., $j$, such that $S_{f,ij}$ is bijective. This implies that $\boldsymbol{S}_{f,i}$ is injective, and therefore it has a left-inverse $\boldsymbol{S}^\dagger_{f,i}$. we apply $\boldsymbol{S}^\dagger_{f,i}$ to both sides of Eq. (31), which gives:

$$Z_i = \boldsymbol{S}^\dagger_{f,i} \mathbf{A}'_i \bar{\boldsymbol{S}}_{f,i}(\tilde{Z}_i). \tag{32}$$

Since $\mathbf{A}'_i$ is a block of an invertible block permutation matrix, $\mathbf{A}_i$ is also an invertible matrix, and therefore $\mathbf{A}'_i$ is a bijective mapping. In addition, since $\tilde{\boldsymbol{S}}_{f,i}$ is injective, $\bar{\boldsymbol{S}}_{f,i}$ is also injective, and therefore the composite map $\boldsymbol{S}^\dagger_{f,i} \mathbf{A}'_i \bar{\boldsymbol{S}}_{f,i} : \mathbb{R} \to \mathbb{R}$ that applies on $\tilde{Z}_i$ is a bijective. This completes the proof that each inferred latent variable $\tilde{Z}_i$ is the bijective transformation of the true latent variable in the case of no noise, where $\boldsymbol{Z} = f^\dagger(\boldsymbol{X})$ are the true latent variables. If noise $\varepsilon$ exists, the posterior distribution of the latent variables can be identified up to an analogous bijective indeterminacy. $\quad\square$

**C.4.4  Consistency**

*Proof.* If the family of the variational posterior $q_\phi(\boldsymbol{Z}|\boldsymbol{X}, Y, T)$ contains the true posterior $p_\theta(\boldsymbol{Z}|\boldsymbol{X}, Y, T)$, then by optimizing the loss of Eq. (9) (with the KL term replaced by the score matching loss defined in Eq. (10)) over its parameter $\phi$, the score matching term will eventually vanish. Therefore, the ELBO term in Eq. (9) will be equal to the log-likelihood. Under this circumstance, CiVAE inherits all the properties of maximum likelihood estimation (MLE). Since the identifiability of CiVAE is guaranteed up to permutation and component-wise bijective transformation of the latent variables, the consistency property of MLE means that the model will converge to the true parameter $\theta^*$ up to such mild indeterminacy of the latent variables in the limit of infinite data. $\quad\square$

**C.5  Proof of Theorem 4.2**

*Proof.* Let $\boldsymbol{C}$ be the true latent confounders and $\tilde{\boldsymbol{C}}$ be the transformed confounders, where the transformation function $f$ is bijective and differentiable *a.e.* Let $f^{-1}$ denote its inverse. The ATE estimator that controls transformed confounders $\tilde{\boldsymbol{C}}$ can be formulated as:

$$DEV(\tilde{\boldsymbol{C}}) = \mathbb{E}_{p(\tilde{\boldsymbol{C}})}[\mathbb{E}[Y|T = 1, \tilde{\boldsymbol{C}} = \tilde{\boldsymbol{c}}] - \mathbb{E}[Y|T = 0, \tilde{\boldsymbol{C}} = \tilde{\boldsymbol{c}}]]. \tag{33}$$

Specifically, for the continuous case where density functions exist, for each term, we have:

$$\mathbb{E}_{p(\tilde{\boldsymbol{C}})}[\mathbb{E}[Y|T = t, \tilde{\boldsymbol{C}} = \tilde{\boldsymbol{c}}]] = \int f_{\tilde{\boldsymbol{C}}}(\tilde{\boldsymbol{c}}) \int y \cdot f_{Y|T,\tilde{\boldsymbol{C}}}(y|t, \tilde{\boldsymbol{c}}) dy d\tilde{\boldsymbol{c}}. \tag{34}$$

For the marginal density $f_{\tilde{\boldsymbol{C}}}(\tilde{\boldsymbol{c}})$, the following equality holds:

$$f_{\tilde{\boldsymbol{C}}}(\tilde{\boldsymbol{c}}) = f_{\boldsymbol{C}}(f^{-1}(\tilde{\boldsymbol{c}}))|J_{f^{-1}}(\tilde{\boldsymbol{c}})| = f_{\boldsymbol{C}}(\boldsymbol{c})|J_{f^{-1}}(\tilde{\boldsymbol{c}})|. \tag{35}$$

As for the conditional density $f_{Y|T,\tilde{\boldsymbol{C}}}(y|t, \tilde{\boldsymbol{c}})$, since $f$ is bijective, according to Eq. (12), we have:

$$f_{Y|T,\tilde{\boldsymbol{C}}}(y|t, \tilde{\boldsymbol{c}}) = f_{Y|T,\boldsymbol{C}}(y|t, \boldsymbol{c}). \tag{36}$$

Combining Eqs. (35) and (36), and given that $d\tilde{\boldsymbol{c}} = |J_f(\boldsymbol{c})| d\boldsymbol{c}$, we have:

$$
\begin{aligned}
(34) &= \int f_{\boldsymbol{C}}(\boldsymbol{c})|\mathbf{J}_{f^{-1}}(\tilde{\boldsymbol{c}})| \int y \cdot f_{Y|T,\boldsymbol{C}}(y|t, \boldsymbol{c}) dy |\mathbf{J}_f(\boldsymbol{c})| d\boldsymbol{c} \\
&= |\mathbf{J}_{f^{-1}}(\tilde{\boldsymbol{c}})| \cdot |\mathbf{J}_f(\boldsymbol{c})| \int f_{\boldsymbol{C}}(\boldsymbol{c}) \int y \cdot f_{Y|T,\boldsymbol{C}}(y|t, \boldsymbol{c}) dy d\boldsymbol{c} \\
&\overset{(a)}{=} \int f_{\boldsymbol{C}}(\boldsymbol{c}) \int y \cdot f_{Y|T,\boldsymbol{C}}(y|t, \boldsymbol{c}) dy d\boldsymbol{c} \\
&= \mathbb{E}_{p(\boldsymbol{C})}[\mathbb{E}[Y|T = t, \boldsymbol{C} = \boldsymbol{c}]],
\end{aligned}
\tag{37}
$$

Table 2: Comparison of CiVAE with baselines when intra-interactions among $M$ exist.

| Dataset | LatentMediator | | LatentCorrelator | | Company (Age) | | Company (Gender) | |
|---------|------|------|------|------|------|------|------|------|
| Method | ATE. | Err. | ATE. | Err. | ATE. | Err. | ATE. | Err. |
| CEVAE | $1.627 \pm 0.549$ | 2.627 | $2.659 \pm 0.302$ | 1.353 | $0.152 \pm 0.027$ | 0.420 | $-0.225 \pm 0.044$ | -0.144 |
| TEDVAE | $1.653 \pm 0.511$ | 2.042 | $2.827 \pm 0.259$ | 1.521 | $0.180 \pm 0.047$ | 0.448 | $-0.189 \pm 0.012$ | -0.108 |
| CiVAE | $\mathbf{-0.350} \pm 0.695$ | **1.785** | $\mathbf{1.785} \pm 0.481$ | **0.479** | $\mathbf{-0.073} \pm 0.101$ | **0.195** | $\mathbf{-0.136} \pm 0.087$ | **-0.055** |
| True ATE | $-1.000 \pm 0.000$ | 0.000 | $1.306 \pm 0.000$ | 0.000 | $-0.268 \pm 0.000$ | 0.000 | $-0.081 \pm 0.000$ | 0.000 |

Table 3: Comparison of CiVAE with baselines when inter-interactions between $C$ and $M$ exist.

| Dataset | LatentMediator | | LatentCorrelator | | Company (Age) | | Company (Gender) | |
|---------|------|------|------|------|------|------|------|------|
| Method | ATE. | Err. | ATE. | Err. | ATE. | Err. | ATE. | Err. |
| CEVAE | $2.070 \pm 0.279$ | 3.070 | $2.831 \pm 0.398$ | 1.831 | $0.094 \pm 0.061$ | 0.362 | $-0.192 \pm 0.015$ | -0.111 |
| TEDVAE | $1.743 \pm 0.307$ | 2.743 | $2.954 \pm 0.763$ | 1.954 | $0.109 \pm 0.116$ | 0.377 | $-0.212 \pm 0.019$ | -0.131 |
| CiVAE | $\mathbf{-0.716} \pm 0.523$ | **0.284** | $\mathbf{1.385} \pm 0.660$ | **0.385** | $\mathbf{-0.041} \pm 0.144$ | **0.227** | $\mathbf{-0.129} \pm 0.064$ | **-0.048** |
| True ATE | $-1.000 \pm 0.000$ | 0.000 | $1.000 \pm 0.000$ | 0.000 | $-0.268 \pm 0.000$ | 0.000 | $-0.081 \pm 0.000$ | 0.000 |

where the term $|J_{f^{-1}}(\tilde{c})| \cdot |J_f(c)|$ vanishes in step (a) as the two factors have the product of one. Therefore, if we plug Eq. (37) into Eq. (33), it leads to the following equality:

$$
\begin{aligned}
DEV(\tilde{C}) &= \mathbb{E}_{p(\tilde{C})}[\mathbb{E}[Y|T=1, \tilde{C}=\tilde{c}] - \mathbb{E}[Y|T=0, \tilde{C}=\tilde{c}]] \\
&= \mathbb{E}_{p(C)}[\mathbb{E}[Y|T=1, C=c] - \mathbb{E}[Y|T=0, C=c]] = DEV(C) = ATE,
\end{aligned}
\tag{38}
$$

where the last step is due to Eq. (2) in Definition 2, which completes our proof that controlling bijectively transformed confounders provides an unbiased estimation of ATE. $\qquad\square$

# D  Extending CiVAE to address Latent Interactions

In this section, we extend CiVAE to more general cases where interactions exist among the latent confounders $C$ and the latent post-treatment variables $M$. Here, we note that the identification of latent confounders $C$ in CiVAE is achieved in two steps. *(i)* CiVAE *individually* identifies latent variables $[C, M]$ that generate $X$ in inferred $Z$ (but which dims of $Z$ correspond to $C$ or $M$ is unknown). *(ii)* pairwise independence test to identify $C$. Since Assumption 2 allows arbitrary dependence among $C$ and $M$, step *(i)* still holds when interactions among $[C, M]$ exist. To distinguish $C$ in these cases, we can use more general causal discovery algorithms, e.g., the PC algorithm [18] in the second step. In this section, we consider two cases of interaction: *(i)* Intra-Interaction among mediators, and *(ii)* Inter-Interaction among mediators and confounders.

## D.1  Intra-Interactions among Latent Mediators

In this subsection, we discuss the case where latent post-treatment variables $M$ interact with each other. Since in this case, $M$ cannot causally influence the latent confounders $C$ (otherwise $C$ will be post-treatment), and the PC algorithm orients edges in causal graphs via colliders, latent confounders can still be identified from the inferred $Z$ as they form colliders with the treatment $T$.

To empirically verify the claim, we extend the simulated datasets described in Section 5.1, where we make *(i)* $T$ directly affects $M_1$, *(ii)* $M_1$ affects $M_2$, and *(iii)* $M_1$, $M_2$ affect $M_3$. The coefficients are randomly sampled from $\mathcal{N}(0, 1/3)$. In step *(ii)*, we use the PC algorithm [18] to identify $C$ from the inferred $Z$. The results in Table 2 demonstrate that the adapted CiVAE is still significantly more robust to latent post-treatment bias compared to CEVAE and TEDVAE, which empirically verify our claim that PC-adapted CiVAE can address the interaction among post-treatment variables.

## D.2  Inter-Interactions between Latent Mediators and Latent Confounders

In this subsection, we discuss another case where inter-interactions exist between latent confounders $C$ and latent post-treatment variables $M$. Since in this case, $M$ still cannot causally influence $C$ (otherwise $C$ will be post-treatment), and the PC algorithm orients edges in causal graph via colliders, latent confounders $C$ can still be identified from $Z$ as they form colliders with the treatment $T$.

To verify the claim, we extend the simulated datasets described in Section 5.1 to allow each latent confounder $C_i \in \mathbb{R}^3$ to determine $M \in \mathbb{R}^3$. The coefficients are randomly sampled from $\mathcal{N}(0, 1/3)$. In step *(ii)*, we use the PC algorithm to identify $C$ from the inferred $Z$. The results in Table 3 demonstrate that the PC-adapted CiVAE is still significantly more robust to latent post-treatment bias compared to CEVAE and TEDVAE, which empirically verify our claim that PC-adapted CiVAE can address the case where inter-interactions exist among latent confounders and post-treatment variables.

