# OpenReview forum: "Causal Effect Estimation with Mixed Latent Confounders and Post-treatment Variables"
_NeurIPS.cc/2024/Conference — Submitted to NeurIPS 2024_

### Official Review · Reviewer_jEuj · 2024-06-16

**Soundness:** 3
**Presentation:** 3
**Contribution:** 3
**Rating:** 6
**Confidence:** 3

**Summary:**

This paper investigates the problem of latent post-treatment bias in causal models where there exists some proxy variables of the latent confounder and post-treatment variables. The authors first derive a general form of latent post-treatment bias which is intractable in most situations (except in special cases such as linear SCM). The authors state that the latent post-treatment bias can be arbitrarily bad for existing proxy-based causal inference methods. They then propose an identifiable VAE-based causal inference algorithm under the assumption that at least one dimension of each sufficient statistic of the latent prior is invertible. The proposed method is evaluated on both synthetic and real-world datasets to demonstrate its causal effect estimation capability with the presence of both latent confounders and post-treatment variables.

**Strengths:**

• Causal reasoning in the context of latent confounder and post-treatment variables is an important topic especially with observational data.

• The authors clearly state the necessary assumptions for the identifiability of true latent variables, and the logic of determining the dimensions of $\boldsymbol{C}$ and $\boldsymbol{M}$ is well presented.

• The paper has a well-established theoretical basis.

**Weaknesses:**

•	For the illustrative example in the introduction, it might be better to explicitly specify what the post-treatment variable is.

•	Other existing works [1-3] on identifying latent confounder/mediators based on the iVAE architecture should also be included in the related work.

•	The role of post-treatment variables $\boldsymbol{M}$ seems to be a bit ambiguous. To be specific, is Theorem 4.1 valid for all types of relationships between $\boldsymbol{M}$ and $Y$?

•	The illustration of (iv) in Assumption 3 is a little confusing, as it assumes one extra degree of freedom on the prior parameters of $\boldsymbol{Z}$ and is critical to the identifiability of $\boldsymbol{Z}$ from $\boldsymbol{X}$. More explanation on this point will be appreciated.

•	The empirical evaluation consists of only one real-world dataset, which somehow limits the applicability of the proposed method.

References:

[1]. Zhou, D., & Wei, X. X. (2020). Learning identifiable and interpretable latent models of high-dimensional neural activity using pi-VAE. Advances in Neural Information Processing Systems, 33, 7234-7247.

[2]. Sorrenson, P., Rother, C., & Köthe, U. (2020). Disentanglement by nonlinear ica with general incompressible-flow networks (gin). arXiv preprint arXiv:2001.04872.

[3]. Jiang, Z., Liu, Y., Klein, M. H., Aloui, A., Ren, Y., Li, K., ... & Carlson, D. (2023). Causal Mediation Analysis with Multi-dimensional and Indirectly Observed Mediators. arXiv preprint arXiv:2306.07918.

**Questions:**

• Is $DEV(\tilde{f}(\boldsymbol{X})) = \mathbb{E}[\mathbb{E}[Y | T = 1, \tilde{f}(\boldsymbol{X})] - \mathbb{E}[Y | T = 0, \tilde{f}(\boldsymbol{X})]]$ in Eq. 4 also based on Lemma 3.1? If yes, it should be explicitly stated.

• In what cases can the bias in Theorem 3.2 be arbitrarily bad besides the causal model assumed by linear SCM in Corollary 3.3?

• What is the rationale behind the simulation procedure of $\boldsymbol{C}$ and $\boldsymbol{M}$ in Eq. 11 if the latent confounder represents “job seniority” as elaborated in the introduction? How do you anticipate the estimation error to change if we increase the complexity of the neural network $NN_f$ in Eq. 11?

**Limitations:**

The authors do not include a paragraph discussing the limitations and potential societal impact of this work.

---

> ### Author Rebuttal · Authors · 2024-08-07
>
> The authors deeply appreciate your insightful comments to make our paper better. We hope that we have addressed your concerns in our responses. If you have further questions, we'd be happy to continue the discussions.
>
> ***Comment 1: Specify the post-treatment variable for the example.***
>
> **Response:**  Thank you for your valuable suggestion. The post-treatment variables in the example are subset of required skills that are causally influenced by the treatment (i.e., switching a job from onsite to online), while influencing people's decisions on applying for the job. For instance, switching to online work might require stronger communication skills, which could affect people's application decisions.
>
> ***Comment 2: Existing works [1-3]  should be included in the related work.***
>
> **Response:** Thank you for pointing out these important works. [1] proposed GIN network in iVAE, which is invertible with volume preservation. [2] generalized [1] by modeling spike outputs with Poisson distribution. [3] adapted iVAE to identify multiple latent mediators from high-dimensional observations. We will include them in the related work.
>
> ***Comment 3: Is Theorem 4.1 valid for all types of relations between 𝑀 and 𝑌?***
>
> **Response:** Thank you for your valuable feedback. Theorem 4.1 is valid for all types of relations between $M$ and $Y$. The reason is that, the non-factorized part in the sufficient statistics of the conditional prior of $Z$ allows for arbitrary dependence between latent variables $Z$ (which includes both $C$ and $M$) and $T, Y$ (see Assumption 2). Therefore, it implicitly allows for any relation between $M$ and $Y$.
>
> ***Comment 4: The illustration of (iv) in Assumption 3 is a little confusing.***
>
> **Response:** Thank you for pointing this out. In this paper, we aim to prove that for CiVAE, if for two $\theta, \tilde{\theta} \in \Theta$ we have $p_{\theta}(X | Y, T)=p_{\tilde{\theta}}(X | Y, T)$, the map from $\tilde{Z}$ to $Z$ is element-wise bijective.  The reason we need $k+1$ $(Y, T)$ points (instead of just $k$) is that, after plugging in $k+1$ $(Y, T)$ points in the equality and taking diff. with the first equation, we can end up with $k$ linearly independent equations (see Eq. (19)), which is necessary to prove the sufficient statistics $S_{f}$ can be identified up to bijective transformation. The details are in Step I of C.4.1.
>
> In addition, Section B.2.3 of [4] shows that if the exponential family parameters $\lambda_{ij}(Y, T)$ are independent, ***(iv)*** can be satisfied with arbitrary $k+1$ different (Y, T) points.
>
> ***Comment 5: The empirical evaluation consists of one real-world dataset.***
>
> **Response:** Thank you for your constructive feedback. We have conducted experiments on the real-world IHDP [5] and the Jobs [6] datasets according to your advice, where we follow the same dataset generation process as the Company dataset to simulate the latent confounders $C$ and the latent post-treatment variables $M$ in the observed covariates $X$. We normalize the outcome $Y$ such that the reported errors become comparable. The results are summarized as follows:
>
> ||IHDP||Jobs||
> |-|-|-|-|-|
> ||ATE|Err $\downarrow$|ATE|Err $\downarrow$|
> |CEVAE|-0.463 ± 0.081|0.787|0.130 ± 0.047|0.525|
> |TEDVAE|-0.317 ± 0.074|0.641|0.185 ± 0.069|0.470|
> |CiVAE|0.178 ± 0.138|0.146|0.602 ± 0.162|-0.053|
> |True ATE| 0.324 ± 0.000 | 0.000 | 0.549 ± 0.000  | 0.000 |
>
> The results further demonsntrate that CiVAE shows more robustness to latent post-treatment bias.
>
> ***Comment 6: Is $DEV(\tilde{f}(X)) = \mathbb{E}[\mathbb{E}[Y|T=1, \tilde{f}(X)] - \mathbb{E}[Y|T=0, \tilde{f}(X)]]$ in Eq. 4 also based on Lemma 3.1?***
>
> **Response:** Thank you for the important question. This step is based on the definition of DEV in Definition 1, where we substitute $X'$ with $\tilde{f}(X)$.
>
> ***Comment 7: In what cases can the bias in Theorem 3.2 be arbitrarily bad?***
>
> **Response:** Thank you for the important question. We provide two linear cases only to intuitively show the latent post-treatment bias (which can be calculated with the coefficients of the linear structural equations). The latent post-treatment bias in the general, nonlinear cases is provided in Eq. (4), which can also be arbitrarily bad, but it is abstract and cannot be further simplified without further assumptions on the causal generation process of the dataset.
>
> ***Comment 8-1: Rationale behind the simulating $C$ and $M$ in Eq. 11 if $C$ represents job seniority***.
>
> **Response:** Thank you for the important question. The example in the introduction aims to provide a concrete example of possible scrambling of latent confounders $C$ (i.e., seniority of the job) and latent post-treatment variables $M$ (i.e., work-mode relevant job skills) in the observed covariates $X$. However, both $C$ and $M$ are difficult to quantify in the Company setting. Therefore, given $(X, T, Y)$, we simulate $C$ and $M$ from scratch, whereas Eq. (11) ensures that the information of the real-world data, i.e., the marginal distribution of the observables, i.e., $(X, T, Y)$, is preserved in the semi-simulated dataset.
>
> ***Comment 8-2: How will estimation error change if we increase the complexity of the neural network $NN_{f}$ in Eq. 11?***
>
> **Response:** Thank you for the important question. As we explained in our response to your comment 8-1, Eq. (11) ensures that the marginal distribution of the observable $(X, T, Y)$ is consistent with the real-world data. A more complicated $NN_{f}$ will make the semi-simulated dataset deviate more from the real-world data (due to overfitting), but it wouldn't affect the estimation step (which is independent of the generation of the semi-simulated dataset).
>
> [4] Variational autoencoders and nonlinear ICA: A unifying framework.
> [5] Bayesian nonparametric modeling for causal inference.
> [6] Evaluating the econometric evaluations of training programs with experimental data.

---

> > ### Comment · Reviewer_jEuj · 2024-08-10
> > **Response to rebuttal**
> >
> > Thank you for your response to my comments, though the negative ATE error on the Jobs dataset looks a bit weird. Also, for the second step in Eq. 4 where you replace $\boldsymbol{X}'$ with $\tilde{f}(\boldsymbol{X})$, I mean this substitution probably needs the injectivity statement in Lemma 3.1.
> >
> > I think the authors' response mostly addresses my concerns, and I'm willing to raise my score.

---

> > > ### Author Response · Authors · 2024-08-10
> > >
> > > **Dear reviewer jEuj,**
> > >
> > > Thank you for the acknowledgment. We are glad to hear that our responses mostly address your concerns. We will try our best to integrate your valuable comments into the paper. Here, we provide responses to your further questions.
> > >
> > > **1.** In the current version, we report the difference between the true ATE and the estimated ATE as the error. We will also provide the absolute difference to make the comparison between different methods more straightforward.
> > >
> > > **2.**  We note that $DEV(X^{\prime})$ is the ATE estimator when controlling an arbitrary variable $X^{\prime}$. $DEV(X^{\prime})$ is defined as: $$DEV(X^{\prime}) := \mathbb{E}_{p(X^{\prime})}[DCEV(X^{\prime})] := \mathbb{E}[\mathbb{E}[Y|T=1, X^{\prime}] - \mathbb{E}[Y|T=0, X^{\prime}]],$$ where ":=" denotes the RHS is the definition of the LHS. Here, $X^{\prime}$ is an arbitrary variable.
> > >
> > > For Eq. (4), we are discussing the bias when estimating the ATE when controlling the variable $\tilde{f}(X)$, i.e., the latent variable inferred from $X$ via $\tilde{f}$. The bias is defined as the difference between the true ATE and the estimated ATE, i.e., $ATE - DEV(\tilde{f}(X))$. Therefore, the second step simply uses the definition of $DEV$, i.e.,  $$ATE - DEV(\tilde{f}(X)) := ATE - \mathbb{E}_{p(\tilde{f}(X))}[DCEV(\tilde{f}(X))] := \mathbb{E}[\mathbb{E}[Y|T=1,\tilde{f}(X)] - \mathbb{E}[Y|T=0, \tilde{f}(X)]],$$
> > > where no derivation is involved in this step. The actual derivations occur in the 3rd and 4th step, where the 4th step uses Lemma 3.1 to remove the injective in the condition.  We will change "=" in the second equation of Eq. (4) to ":=" to avoid confusion. Thank you again for raising this important question to make our paper clearer.
> > >
> > > Best,
> > > Authors

---

> > > > ### Comment · Reviewer_jEuj · 2024-08-11
> > > > **Response to author's comment**
> > > >
> > > > Thank you for your clarification. I have no further questions at this point.

---

> > > > > ### Author Response · Authors · 2024-08-11
> > > > >
> > > > > **Dear Reviewer jEuj,**
> > > > >
> > > > > Thank you for the acknowledgment. We are committed to integrating your valuable comments into the paper. Thank you again for your time and efforts.
> > > > >
> > > > > Best,
> > > > > Authors

---

### Official Review · Reviewer_FDXX · 2024-07-07

**Soundness:** 3
**Presentation:** 3
**Contribution:** 3
**Rating:** 6
**Confidence:** 2

**Summary:**

The authors deal with latent post-treatment bias for proxy-based methods which are employed for causal effect estimation.
They show that post-treatment variables can be latent and mixed into the observed covariates along with the latent confounders.
The authors transform the confounder-identifiability problem into a tractable pair-wise conditional independence test problem.
They prove that the latent confounders and latent post-treatment variables can be identified up to bijective transformations. Finally, they provide experimental analysis for their approach.

**Strengths:**

The paper deals with a very interesting problem.	 The proposed method appeared to be theoretically robust. The method is evaluated with proper experimental analysis on synthetic and real-world datasets and compared with multiple benchmarks.

**Weaknesses:**

Here I provide some weaknesses of the paper:
* Bi-directed edges in Figure 1 are not defined properly.
* Do-operator in equation 3 is not defined in detail.
* Assumptions in Assumption 2 should be described in more detail.
* The proposed method seems to depend on a lot of assumptions. Assumptions 1,2,3 each contain multiple assumptions. The authors should explain how their assumptions hold for the real-world scenarios they considered in their experiment section.

**Questions:**

Here I provide some questions:
* Why do the authors assume that the models can recover the true latent space up to invertible transformation (line 151) ? How realistic is that assumption?
* Do the proxy-based methods claim to perform well for the causal graphs in Fig 1c?
* How do these assumptions hold when X is high-dimensional?
* What values of K_C and K_M are considered?

**Limitations:**

The authors discussed a very few limitations of their paper but more discussion should be done.

---

> ### Author Rebuttal · Authors · 2024-08-07
>
> The authors deeply appreciate your insightful comments to make our paper better. We hope that we have addressed your concerns in our responses. If you have further questions, we'd be happy to continue the discussions.
>
> ***Comment 1: Bi-directed edges in Figure 1 are not defined properly.***
>
> **Response:** Thank you for pointing this out. Bi-directed edges in Fig. 1 means arbitrary causal relationship between each of the post-treatment variable $M_{i}$ and the outcome $Y$. This will be clarified in the revised paper.
>
> ***Comment 2: Do-operator in equation 3 is not defined.***
>
> **Response:** Thank you for pointing this out. The do-operator represents an intervention in the causal model, where $\mathbb{E}[Y|do(T=t)]$ represents the expected value of $Y$ if we were to intervene and set the treatment $T$ to value $t$ for the entire population, regardless of the natural causes of $T$. We will add this explanation to the paper for clarity.
>
> ***Comment 3:  Assumption 2 should be described in more detail.***
>
> **Response:** Thanks for the constructive suggestion! The assumption states the mild condition the prior of $Z$, i.e., $P(Z|Y, T)$, should satisfy for individual and bijective identification of $Z$ from the observables $(X, Y, T)$. Specifically, it assumes that $P(Z|Y, T)$ belongs to a general exponential family with two-part sufficient statistics $S(Z)$: ***(i)*** A factorized part $S_f(Z)$, where each component $S_i(Z_i)$ has at least one invertible dimension. This ensures individual/bijective identifiability of latent variables. ***(ii)*** A non-factorized part $S_{nf}(Z)$ modeled by a ReLU deep neural network, which allows complex dependencies among latent variables. We will add the above explanation to the paper.
>
> ***Comment 4: How the assumptions hold for the real-world scenarios in the experiment section?***
>
> **Response:** Thank you for your insightful suggestion. We provide empirical justification for the three assumptions as follows:
>
> For Assumption 1, since the dimension of the observed covariates $X$, i.e., $K_{X}$, is larger than the dimension of the latent variables $Z=[C, M]$, i.e., $K_{Z} = K_{C} + K_{M}$, it would be likely that $f$ is injective or very close to injective due to the low probability that two distant points in the low dimensional latent space are mapped by $f$ to the same point in the high-dimensional $X$ space.
>
> Assumption 2, i.e., the conditional prior of latent variables following a general exponential family, would be a reasonable approximation of the true prior, as general exponential family includes the most commonly used distributions, and the non-factorized part of the sufficient statistics parameterized by a ReLU deep neural network allows complex (conditional) dependence among the latent variables.
>
> Assumption 3 ensures that the dataset and model class we choose allow the identification, where ***(i)*** states that the noise distribution should not be degenerative, which depends on the dataset quality. ***(ii)***, ***(iii)*** can be trivially satisfied by neural networks. For ***(iv)***,  Section B.2.3 [1] shows that if the exponential family parameters $\lambda_{ij}(Y, T)$ are independent, ***(iv)*** can be satisfied with arbitrary $k+1$ different $(Y, T)$ points. This can be satisfied by most exponential family distributions.
>
> ***Comment 5:  Why do the authors assume that the models can recover the true latent space up to invertible transformation (line 151)?***
>
> **Response:** Thanks for the valuable feedback. We want to clarify that this assumption is made only for existing proxy-based methods, not for our proposed CiVAE. This is actually the most optimistic assumption we could make for these methods as they exactly recover the latent space. We show that even under this optimistic assumption, the ATE/CATE estimation is still arbitrarily biased when latent post-treatment variables are present.
>
> ***Comment 6: Do the proxy-based methods claim to perform well for the causal graphs in Fig 1c?***
>
> **Response:** Thanks for the valuable feedback. Most proxy-based methods do not explicitly address the causal graph in Fig 1c. They typically rely on the strong ignorability assumption, which implies both that ***(i)*** all confounders are captured by observed covariates, and ***(ii)*** no post-treatment variables are included. However, these methods often focus more on the first implication and ignore the potential presence of post-treatment variables in the proxies (which leads to violation of ***(ii)***). This can lead to biased ATE/CATE estimates when latent post-treatment variables are mixed with confounders in the observed covariates, and motivates us to design CiVAE to address the latent post-treatment bias.
>
> ***Comment 7: How do these assumptions hold when $X$ is high-dimensional?***
>
> **Response:** Thanks for raising this important point. Assumption 1 (noisy-injectivity) implies that the dimension of $X$ is **larger** than or equal to the latent space, which is typically satisfied when $X$ is high-dimensional. Assumption 2 puts a general prior on the latent variables, whereas Assumption 3 contains standard regularity conditions, which are both irrelevant to the dimensionality of $X$. Therefore, all three assumptions in this paper hold for high-dimensional covariates $X$.
>
> ***Comment 8: What values of K_C and K_M are considered?***
>
> **Response:** Thanks for the valuable feedback. In our experiments, we considered various combinations of $K_{C}$ and $K_{M}$: For the simulated datasets, we empirically set $K_C = 3$ and $K_M = 3$. For the real-world Company dataset, we empirically set $K_{C} = 5$ and $K_{M} = 3$. Additionally, in Section 5.3, for the Company dataset, we conducted a sensitivity analysis where we varied the ratio of $K_{C}$ to $K_{M}$ from $\\{2:6, 3:5, 4:4, 5:3, 6:2\\}$. This analysis demonstrates the robustness of CiVAE under different latent variable configurations.
>
> [1] Variational autoencoders and nonlinear ICA: A unifying framework.

---

### Official Review · Reviewer_kHjK · 2024-07-12

**Soundness:** 3
**Presentation:** 2
**Contribution:** 2
**Rating:** 5
**Confidence:** 3

**Summary:**

This paper addresses the challenge of causal inference with observational data, particularly when direct measurement of confounders is infeasible. The authors propose a new method, Confounder-identifiable Variational Autoencoder (CiVAE), to mitigate post-treatment bias using observed proxies for both latent confounders and latent post-treatment variables. The paper provides a theoretical analysis under specific assumptions and validates the proposed approach through experiments on both simulated and real-world datasets.

**Strengths:**

* The paper investigates a critical question concerning the mitigation of post-treatment bias, which is essential in various practical scenarios.
* The ideas presented in the paper are clear and easy to follow, and the theoretical analysis is well-established.

**Weaknesses:**

* In practical scenarios, interactions among latent factors are often present and can significantly impact the estimation. It would be beneficial if the authors could elaborate on how their method addresses these interactions and whether there are any theoretical guarantees regarding their handling in the proposed approach.

* The theoretical guarantees rely on strong assumptions, and the assumptions are hard to verify in practice. In assumption 1, the paper assumes an injective function of latent confounders and latent post-treatment variables into the observed proxy. This is a strong assumption,  and it will be much harder to meet the assumption in general when the function is nonlinear. The specific setup with strong assumptions limits the practical applicability of the proposed approach. It would be helpful if the authors could provide examples where these assumptions hold and demonstrate how they can be verified.

* The experiment lacks sufficient details on setup and implementation. Could the authors provide more specific information to enhance understanding of the empirical results?

**Questions:**

See Weaknesses.

**Limitations:**

* The proposed method relies on very strong assumptions to ensure identifiability, which can be challenging to verify in practical applications.

---

> ### Author Rebuttal · Authors · 2024-08-07
>
> The authors deeply appreciate your insightful comments to make our paper better. We hope that we have addressed your concerns in our responses. If you have further questions, we'd be happy to continue the discussions.
>
> ***Comment 1: How CiVAE addresses interactions among latent variables and theoretical guarantees?***
>
> **Response:** Thank you for raising this important point. We have extended our analysis to interactions among latent variables in Section 4 of the Appendix. Specifically, we consider two cases: ***(i)*** Intra-interactions among latent mediators $M$ and ***(ii)*** Inter-interactions between $M$ and latent confounders $C$.
>
> **Theoretically**, the inferred latent variables $\hat{Z}$ via Eq. (10) still individually identify the true latent variables, i.e., $[C, M]$, up to bijective map, as Assumption 2 allows arbitrary (conditional) dependence among latent variables. When interactions exist, we can use more general causal discovery methods, e.g., the PC algorithm, to further identify the latent confounders $C$ in $\hat{Z}$. The reason is that, since latent post-treatment variables $M$ cannot causally influence $C$ (otherwise $C$ will be post-treatment), and the PC algorithm orients edges in the causal graph via colliders, latent confounders $C$ can be properly oriented by the PC algorithm as they form colliders with $T$ and therefore be identified from $\hat{Z}$.
>
> **Empirically**,  we simulate two datasets according to the above two cases, and show that CiVAE can be adapted to handle these interactions by adopting the PC algorithm in the second step of confounder identification. Tables 1 and 2 in the Appendix demonstrate that the adapted CiVAE remains more robust to latent post-treatment bias compared to baselines even when interactions exist among latent variables.
>
> ***Comment 2-1: Assuming an injective function of latent confounders and post-treatment variables into the observed proxy is strong, and it will be harder to meet the assumption in general when the function is nonlinear.***
>
> **Response:** Thank you for your insightful comments. The proposed noisy-injectivity assumption is weaker than a strict injective assumption, as it allows the map from latent variables $Z$ (including latent confounders $C$ and latent post-treatment variables $M$) to the **observed** covariates $X$ to be non-injective due to the presence of noise.
>
> In addition, since the dimension of the observed covariates $X$, i.e., $K_{X}$, is larger than the dimension of the latent variables $Z=[C, M]$, i.e., $K_{Z} = K_{C} + K_{M}$, it would be likely that $f$ is injective or very close to injective in practice due to the low probability that two distant points in the low dimensional latent space are mapped by $f$ to the same point in the high-dimensional $X$ space. We will clarify the above points in the revised paper.
>
> ***Comment 2-2: It would be helpful if the authors could provide examples where assumptions hold and demonstrate how they can be verified.***
>
> Thank you for your insightful comments. For the remaining two assumptions, we provide further discussion as follows:
>
> Assumption 2, i.e., the conditional prior of latent variables following a general exponential family, would be a reasonable approximation of the true prior. The reason is that, the non-factorized part of the sufficient statistics of general exponential family defined in Eq. (7) is parameterized by a ReLU deep neural network, which allows complex (conditional) dependence among the latent variables.
>
> Assumption 3 ensures that the dataset and model class we choose allow the identification. Specifically, ***(i)*** denotes that the noise distribution should not be degenerative, which depends on the dataset quality. ***(ii)***, ***(iii)*** can be trivially satisfied by neural networks. For ***(iv)***,  Section B.2.3 of [1] shows that if the factorized part of the exponential family parameters $\lambda_{ij}(Y, T)$ are independent (which is very weak), ***(iv)*** can be satisfied with **arbitrary** $k+1$ different $(Y, T)$ points.
>
> [1] Variational autoencoders and nonlinear ICA: A unifying framework.
>
> ***Comment 3: The experiment lacks sufficient details on setup and implementation. Could the authors provide more specific information to enhance understanding of the empirical results?***
>
> **Response:** Thank you for your constructive feedback. The detailed setup and implementation are summarized as follows:
>
> For the simulated datasets, we empirically set the dimension of the latent confounders and the latent post-treatment variables to $K_{C}=3$ and $K_{M}=3$, which leads to $K_{Z}=K_{C} + K_{M}=6$. The dimension of the observed covariates is set to $K_{X}=20$. The dataset generation process for both the ***mixedLatentMediator*** and ***mixedLatentCorrelator*** cases have been formulated in the paper. For CiVAE, the inference network $q_{\phi}(Z|X, Y, T)$ is implemented as an MLP with one hidden layer with hidden dimension $K_{H}=K_{Z}$. For the prior network $p_{S, \lambda}(Z|Y, T)$: for the factorized part, we implement $S_{f}(Z) = [Z, Z^{2}]$ and implement $\lambda_{f}(Y, T)$ as a dense layer of $\mathbb{R}^{2} \rightarrow \mathbb{R}^{2 \times K_{Z}}$; for the non-factorized part, we implement $S_{nf}(Z)$ as a ReLU neural network with hidden dimension of $K_{H}=K_{Z}$ and output dimension of 1, and implement $\lambda_{nf}(Y, T)$ as a dense layer of $\mathbb{R}^{2 \rightarrow 1}$. We train the model according to Eq. (10) for ten epochs, conduct ten random runs of the experiment, and report the average and standard deviation.
>
> For the real-world dataset, we select 52 most common job skills as $X$ (which leads to $K_{X}=52$). We set the dimension of the latent space, i.e., $K_{Z} = 8$, and vary the ratio $K_{C} : K_{M}$ from $2:6$ to $6:2$ and plot the results in Fig. 3. The implementation and training of CiVAE follow the same setting as the simulated datasets. We will include the above details in the revised paper.

---

> > ### Comment · Reviewer_kHjK · 2024-08-13
> >
> > I thank the authors for providing detailed responses, and these partially address the concerns. I will keep my rating.

---

> > > ### Author Response · Authors · 2024-08-13
> > >
> > > **Dear Reviewer kHjK,**
> > >
> > > Thank you for the acknowledgment. We are glad that our responses partially address your concerns. We will take your remaining comments seriously and further polish the paper, and we are committed to integrating your valuable comments into the paper. Thank you again for your time and efforts.
> > >
> > > Best,
> > > Authors

---

### Official Review · Reviewer_VF9h · 2024-07-12

**Soundness:** 3
**Presentation:** 3
**Contribution:** 3
**Rating:** 7
**Confidence:** 2

**Summary:**

In this paper, the authors investigated the issue of latent post-treatment bias in causal inference from observational data. They showed that estimator of existing proxy-of-confounder-based methd, i.e., DEV (f(X)), is an arbitrarily biased estimator of the Average Treatment Effect (ATE), when the selected proxy of confounders X accidentally mixes in latent post-treatment variables (Theorem 3.2). To address this issue, they proposed the Confounder-identifiable VAE (CiVAE), which identifies latent confounders up to bijective transformations under a mild assumption regarding the prior of latent factors. They showed that controlling for latent confounders inferred by CiVAE can provide an unbiased estimation of the ATE. Experiments on both simulated and real-world datasets demonstrate that CiVAE exhibits superior robustness to latent post-treatment bias compared to state-of-the-art methods.

**Strengths:**

Being able to recover latent variables (cofounders, post-treatment variables, or others) from observations is challenging and important. Ignoring latent variables or assuming non-existence of latent variables is unrealistic and can lead to the wrong conclusion and decisions. The authors further motivated the importance of recovering latent cofounders, post-treatment variables and the consequence of not doing so  (Theorem 3.2). The solution provided shows originality and quality.

**Weaknesses:**

The presentation can be improved.

**Questions:**

Is Fig. 1(c) general enough? It assumes that all latent variables are either confounders or post-treatment variables. However, there can be other types of latent variables, such as:
1. Pre-treatment Variables: These latent variables influence the treatment (T) but do not directly affect the outcome (Y) or the additional observation (X). They exist before the treatment is applied and can introduce selection bias.
2. Latent Interaction Variables: These latent variables interact with the treatment (T) to influence the outcome (Y). They are not confounders because they do not influence the treatment directly, nor are they post-treatment variables.
3. Latent Mediator Variables: These latent variables mediate the effect of the treatment (T) on the outcome (Y) and are not directly observed.
4. Latent Variables Influencing Both Pre-treatment and Post-treatment States: These latent variables influence the state of the system both before and after the treatment but do not fit the typical definition of confounders or post-treatment variables. For example, a latent mental state might affect both a person's initial willingness to undergo treatment and their behavior or responses post-treatment.

Can the proposed method handle these types too (with some extension), or some of the types are quite disruptive to the proposed methodology?

**Limitations:**

n.a.

---

> ### Author Rebuttal · Authors · 2024-08-07
>
> The authors deeply appreciate your insightful comments to make our paper better. We hope that we have addressed your concerns in our responses. If you have further questions, we'd be happy to continue the discussions.
>
> ***Comment 1: How does CiVAE (with possible extension) address other types of latent variables?***
>
> **Response:** Thank you for the insightful comments. It would indeed be interesting to discuss the behavior and possible extension of CiVAE when different types of latent variables $W$ are scrambled into the observed covariates $X$ alongside the latent confounders $C$ and the latent post-treatment variables $M$.
>
> First, from Assumption 2, we know that CiVAE allows arbitrary (conditional) dependence among latent variables $Z$ that generates $X$ to **individually** and **bijectively** identify them from the observables $(X, Y, T)$. Therefore, ***regardless of the type of*** $W$, if we denote the inferred latent variables as $\hat{Z}$, we have $\hat{Z}\_{i} \in \\{f\_{k}(W\_{k}), f\_{k'}(C\_{k'}),  f\_{k''}(M\_{k''})\\}$, where $f$ is a bijective function. However, for each $i$, whether $\hat{Z}_{i}$ corresponds to type $W$, $C$ or $M$ is unknown.
>
> Therefore, if only $C$ and $M$ exist, to further distinguish $C$ from $M$ in $\hat{Z}$, since $C$ are pre-treatment and $M$ are post-treatment, a clever strategy is to select variables in $\hat{Z}$ where independence increases after conditioning on $T$, as only $C$ form V-structure with $T$ (i.e., $C_{i} \rightarrow T \leftarrow C_{j}$), where their dependence increases after conditioning on $T$. In contrast, $C$ and $M$ form chain structure with $T$ (i.e., $C_{i} \rightarrow T \rightarrow M_{j}$), and $M$ form fork structure with $T$ (i.e., $M_{i} \leftarrow T \rightarrow M_{j}$), where dependence decreases after conditioning on $T$.
>
> We can use similar logic to reason with the case when different types of $W$ exist.
>
> ***Case 1. Pre-treatment variables that do not direct influence the outcome.***
>
> If $W$ are pre-treatment variables, since they causally influence the treatment $T$,  they still form V-structure with $T$, and therefore CiVAE will identify them in $\hat{Z}$ and include them in the control set after the pair-wise independence test.
>
> Here, we need to further divide the pre-treatments $W$ into two cases.
>
> The first case is that $W$ are correlated with $Y$. In this case, controlling the identified $W$ can reduce both confounding bias and variance.
>
> Another case is that $W$ are not-correlated with $Y$. In this case, controlling $W$ is still unbiased (which achieves the main purpose of removing latent post-treatment bias of the paper), but the estimation variance could increase. A trivial extension of CiVAE to address this issue is to conduct another round of independence test among the identified confounders $\hat{C}$ (with and without the outcome $Y$ as the condition) and keep the pairs in $\hat{C}$ where dependence increases after conditioning on $Y$ (as true confounders form V-structure with $Y$). The discussion will be included in the revised paper.
>
> ***Case 2.   Latent interaction variables.***
>
> The case where $W$ are latent interaction variables is more complicated, as the relation between $W$ and the treatment $T$ is undetermined. If each $W_{i}$ is confounded with $T$ via an independent unobserved confounder $U_{i}$, $W$ have the following relationship with $T$, i.e., $W\_{i} \leftarrow U\_{i} \rightarrow T \leftarrow U\_{j} \rightarrow W_{j}$. Since the dependence among $W$ will increase after conditioning on $T$, $W$ will be included in the control set. However, if $W$ is confounded with $T$ via a shared confounder $U$, the relation becomes $T \leftarrow U \rightarrow \{W\_{i}, W\_{j}\}$, controlling $T$ would probably decrease the dependence (as $T$ contains the confounder information). In this case, $W$ won't be included in the control set.
>
> However, regardless of whether $W$ are included in the control set, CiVAE remains unbiased, because $W$ do not influence the identification of confounders $C$. In addition, $W$ are still pre-treatment, such that no post-treatment bias can be introduced in the ATE/CATE estimation.
>
> ***Case 3. latent mediator variables.***
>
> If $W$ are latent mediators, $W$ is a special case of post-treatment variables $M$. Since $W$ form the fork structure with the treatment $T$ (i.e., $W\_{i} \leftarrow T \rightarrow W\_{j}$), their dependence will decrease after conditioning on $T$, and therefore they will be successfully excluded from the control set to eliminate latent post-treatment bias.
>
> ***Case 4. latent variables influencing both pre-treatment and post-treatment states?***
>
> If $W$ are latent variables that influence both pre-treatment and post-treatment states, since $W$ still forms the fork structure with the treatment $T$ (i.e., $W\_{i} \leftarrow T \rightarrow W\_{j}$), their dependence will decrease after conditioning on $T$, and therefore they will be successfully excluded from the control set to eliminate latent post-treatment bias.

---

> > ### Comment · Reviewer_VF9h · 2024-08-09
> >
> > Thank you. I've read your rebuttal, responses, and the other reviews. I will keep an eye on the reviewers' discussion phase, if there is one.

---

> ### Author Response · Authors · 2024-08-09
>
> **Dear reviewer VF9h,**
>
> Thank you for the acknowledgment. We will try our best to integrate your valuable comments into the paper. Thank you again for your time and efforts.
>
> Best,
> Authors

---

### Decision · Program_Chairs · 2024-09-25

**Decision:**

Reject

**Comment:**

This paper addresses causal effect estimation with unobserved confounders, focusing on recovering confounding information from auxiliary proxy variables. Specifically, it tackles the challenge of proxies that capture information about both unobserved confounders and latent post-treatment variables, which can introduce post-treatment bias. The paper proposes a VAE approach to individually recover latent confounders and post-treatment variables up to bijective transformations. It then aims to disentangle these components and adjust for latent confounders in causal effect estimation.

While the reviews are overall positive, some concerns were raised, including the plausibility of stringent assumptions, handling of general confounders and post-treatment variables, and the presentation of experimental results. The rebuttals partially addressed these concerns, but fully resolving the issues, especially regarding general confounders and post-treatment variables, requires significant work. The paper emphasizes the VAE component and briefly mentions using pairwise independence tests to disentangle confounders and post-treatment variables under a restrictive independence assumption. The appendix offers brief discussions on more general cases without independence restrictions. However, this is a very cursory treatment of the disentanglement procedure, one key component that differentiates the work from existing literature. A deeper study of the disentanglement process is needed, including clear procedures, validity explanations, and validation through well-designed ablation studies.

During the AC-reviewer discussion, another crucial question arose: Is the problem studied practically relevant? It is unclear when the problem with post-treatment proxies would be encountered in real scenarios. Since confounders are pre-treatment, it's generally more reasonable to use pre-treatment information. The paper lacks a discussion on the practical relevance of the problem, providing only a simplified and somewhat unrealistic example in the introduction that may represent only a narrow use case. Additionally, the example involving job mode switch and applicants' age is confusing, as age is an exogenous attribute likely influencing decisions rather than the other way around, and variables like seniority and skills are also ambiguous.

Given these issues, I recommend rejection at this stage.